# Regret Minimization for Reinforcement Learning with Vectorial Feedback and Complex Objectives

**Wang Chi Cheung**
Department of Industrial Systems Engineering and Management
National University of Singapore
`isecwc@nus.edu.sg`

## Abstract

We consider an agent who is involved in an online Markov Decision Process, and receives a vector of outcomes every round. The agent optimizes an aggregate reward function on the multi-dimensional outcomes. Due to state transitions, it is challenging to balance the contribution from each dimension for achieving near-optimality. Contrary to the single objective case, stationary policies are generally sub-optimal. We propose a no-regret algorithm based on the Frank-Wolfe algorithm (Frank and Wolfe 1956, Agrawal and Devanur 2014) , UCRL2 (Jaksch et al. 2010), as well as a crucial and novel Gradient Threshold Procedure (GTP). GTP involves carefully delaying gradient updates, and returns a non-stationary policy that diversifies the outcomes for optimizing the aggregate reward.

## 1 Introduction

Markov Decision Processes (MDPs) model sequential optimization problems with changes in the state of the underlying environment. At each time, an agent performs an action, contingent upon the current state. Influenced by the present state and action, the agent transits to another state and receives some form of feedback. Typically, the feedback is a scalar reward, and the agent aims to maximize the total reward. Nevertheless, in many settings, the feedback is a vector of multiple outcomes, and the agent's goal depend on each of these outcomes. Moreover, the underlying MDP model is usually not known to the agent, and is to be learned on-the-fly. Motivated by these situations, we consider the Complex-Objective Online MDP (CO-OMDP) problem, which maximizes an aggregate function on the average vectorial outcome.

Solving the CO-OMDP problem requires overcoming the following subtle challenges. To maximize the aggregate function, an agent has to balance the contributions from the outcomes' different components by alternating among different actions, which are generally associated with different states. Consequently, the agent has to traverse the state space, which could require visiting sub-optimal states that do not contribute to the maximization of the aggregate function. Altogether, the maximization can be hindered by undesirable state transitions, which is worsened by the agent's model uncertainty.

We overcome the mentioned challenges by proposing TFW-UCRL2, a near-optimal online algorithm for the CO-OMDP problem. The algorithm is built upon the Frank-Wolfe algorithm (FW) [21, 2], UCRL2 [28], as well as our novel Gradient Threshold Procedure (GTP). FW balances the objectives by scalarizing the outcomes, and UCRL2 solves scalarized online MDP problems under model uncertainty. However, FW and UCRL2 are not enough for overcoming the challenges in balancing the outcomes while avoiding sub-optimal states. GTP overcomes the challenges by judiciously delaying the gradient updates in FW. The procedure approximately maintains the balancing effect by FW, while limits the visits to sub-optimal states by switching among different stationary policies adaptively and infrequently, despite the model uncertainty.

**Related Literature.** The CO-OMDP problem is closely related to the Bandits with global concave Rewards problem (BwR) [2] and the Scalar-Obejctive Online MDPs problem (SO-OMDP) [28]. BwR concerns maximizing aggregate functions on vectorial feedback in stochastic bandit settings. BwR is first studied by [2], who solve BwR by a synergy of online convex optimization and upper confidence bound (UCB) algorithms. Subsequently, BwR is studied under various stochastic bandit settings and reward functions [4, 17, 14]. BwR is closely related to Bandit with Knapsacks problem (BwK), which models stochastic bandit problems with resource constraints. BwK is first studied by [10], and is subsequently studied in various settings [2, 11, 3, 20]. The models for BwR and BwK assume i.i.d. outcomes across time, while in an MDP setting the outcome distribution changes endogenously according to the state transition.

The adversarial BwK problem is recently studied in [27]. Among other results, they show that no online algorithm can achieve an expected value of $\Omega(1/\log T)$ times the offline optimum, where $T$ is the number of time steps. Our positive results on CO-OMDPs walk a fine line between the negative results for adversarial BwK and the positive results for stochastic BwR and BwK. Online optimization problems with global reward functions are studied in adversarial settings with full feedback [19, 9].

The SO-OMDP problem is first studied by [8, 28] for communicating MDPs. Subsequently, the problem is studied under the more general cases of weakly communicating MDPs [13, 23] and non-communicating MDPs [22]. Posterior sampling algorithms for the SO-OMDP problem are proposed and analyzed [5, 37]. The SO-OMDP problem is also studied under certain mixing time assumptions on all stationary policies [36]. The SO-OMDP problem assumes scalar rewards, but do not incorporate multi-objective optimization. For a review on MDPs, please consult [38, 15].

Reinforcement Learning (RL) with vectorial feedback and aggregate functions are studied in the discounted-reward setting [24, 6, 12, 43, 1, 25, 42, 29, 30, 33] and the average-reward [6, 32, 41, 40] setting. We study the latter with an online model under model uncertainty. Our work shows non-asymptotic convergence to the optimum, which differs from [6, 32] who show asymptotic convergence. Tarbouriech and Lazaric [41] study an online model for state space exploration, and achieve a non-asymptotic convergence to the optimum. In addition to the choice of the aggregate functions, our work differs from [41] in two aspects. First, the transition kernel is assumed to be known in [41], whereas the kernel is not known in our model. Second, the reachability assumption of unichain MDPs is made in [41], while we make the much weaker assumption of communicating MDPs. More recently, online MDPs with adversarially chosen aggregate functions are studied by [40]. The model in [40] is episodic, where the state is reset to a fixed state at the end of an episode (involving a fixed number of steps). In contrast, our setting does not involve any state reset. In [40] the aggregate function is applied only on the trajectory in each episode, whereas in our setting the aggregate function is applied on the trajectory across the whole horizon. Finally, we point out that a substantial generalization of the current paper has been put forth in [18].

In the discounted reward setting, multi-objective optimization are studied in [24, 6, 12, 43]. Many recent works study the discounted-reward setting with resource constraints [1, 42, 29, 33]. Numerous recent research works focus on state space exploration problems [25, 30] in the discounted-reward setting. Constrained MDPs are reviewed in [6], and multi-objective RL is surveyed in [39, 31].

## 2 Problem Definition of CO-OMDP

A CO-OMDP instance is specified as $(\mathcal{S}, s_1, \mathcal{A}, p, \mathcal{V}, g)$. The set $\mathcal{S}$ is a finite state space, and $s_1 \in \mathcal{S}$ is the starting state. The collection $\mathcal{A} = \{\mathcal{A}_s\}_{s \in \mathcal{S}}$ contains a finite set of actions $\mathcal{A}_s$ for each state $s$. We say $(s, a)$ is a state-action pair iff $s \in \mathcal{S}, a \in \mathcal{A}_s$. The collections $p = \{p(\cdot|s,a)\}_{s \in \mathcal{S}, a \in \mathcal{A}_s}$ is the transition kernel, and the collection $\mathcal{V} = \{\mathcal{V}(s,a)\}_{s \in \mathcal{S}, a \in \mathcal{A}_s}$ governs the vectorial outcomes. When the agent chooses action $a \in \mathcal{A}_s$ at state $s$, her subsequent state $s'$ is distributed as $p(\cdot|s,a) \in \Delta^{\mathcal{S}}$. She receives a stochastic vectorial outcome $V(s,a) \in [0,1]^K$, distributed as $\mathcal{V}(s,a)$, and has mean $\mathbb{E}[V(s,a)] = v(s,a) = (v_k(s,a))_{k=1}^K$. We emphasize that $s', V_1(s,a), \ldots, V_K(s,a)$ can be arbitrarily correlated. We focus on the following reward function $g : [0,1]^K \to \mathbb{R}_{\geq 0}$, which is parameterized by $L_0 \in \mathbb{R}_{\geq 0}, L_1, \ldots, L_K \in \mathbb{R}$, and a convex compact set $U \subseteq [0,1]^K$:

$$g(w) := \frac{1}{K} \cdot \left[ \sum_{k=1}^K L_k w_k - \frac{L_0}{2} \min_{u \in U} \left\{ \sum_{k=1}^K (w_k - u_k)^2 \right\} \right]. \tag{1}$$

The function $g$ is concave (see Appendix B.1), and is to be maximized.

**Dynamics.** An agent, who faces an CO-OMDP instance $\mathcal{M} = (\mathcal{S}, s_1, \mathcal{A}, p, \mathcal{V}, g)$, starts at state $s_1 \in \mathcal{S}$. At time $t$, three events happen. First, the agent observes his current state $s_t$. Second, she takes an action $a_t \in \mathcal{A}_{s_t}$. Third, she transits to another state $s_{t+1} \sim p(\cdot|s_t, a_t)$, and receives the vectorial outocme $V_t(s_t, a_t) \sim \mathcal{V}(s_t, a_t)$. Both $s_{t+1}$ and $V_t(s_t, a_t)$ are observed by the agent. The whole dynamics result in a controlled Markov process $\{s_t, a_t, V_t(s_t, a_t)\}_{t=1}^{\infty}$. Conditioned on $(s_t, a_t)$, the random variable pair $(s_{t+1}, V_t(s_t, a_t))$ is independent of $H_{t-1}$.

In the second event, the choice of $a_t$ is based on a *non-anticipatory* policy. The choice only depends on the current state $s_t$ and the previous observations $H_{t-1} := \{s_q, a_q, V_q(s_q, a_q)\}_{q=1}^{t-1}$. When $a_t$ only depends on $s_t$, but not on $H_{t-1}$, the corresponding non-anticipatory policy is said to be *stationary*.

**Objective.** The CO-OMDP instance $\mathcal{M}$ is latent. While the agent knows $\mathcal{S}, s_1, \mathcal{A}, g$, she does not know $v, p$. To state the objective, define $\bar{V}_{1:t} := \frac{1}{t} \sum_{q=1}^{t} V_q(s_q, a_q)$. For any horizon $T$ not known *a priori*, the agent aims to maximize $g(\bar{V}_{1:T})$, by selecting actions $a_1, \ldots, a_T$ with a non-anticipatory policy. Denote $\bar{V}_{1:T,k}$ as the $k$-component of the time average vector $\bar{V}_{1:T}$. CO-OMDPs capture the following problems:

*Multi-Objective Optimization.* Consider maximizing the scalar function $\sum_{k=1}^{K} L_k \bar{V}_{1:T,k}$, while trying to meet the Key Performance Index (KPI) requirement $\bar{V}_{1:T,k} \geq \rho_k$ for each $k \in \{1, \ldots, K\}$. The vector $\rho = (\rho_k)_{k=1}^{K} \in [0,1]^K$ comprises the pre-determined KPI targets for the $K$ objectives $\{\bar{V}_{1:T,k}\}_{k=1}^{K}$. The task can be modelled as a CO-OMDP problem, by setting $L_0 \geq 0$, and $U = \{w : w_k \geq \rho_k \, \forall 1 \leq k \leq K\}$. By putting $\rho_k = 1$ and $L_k = 0$ for each $k$, any maximizer of $g(\bar{V}_{1:T})$ is *Pareto-optimal* for the simultaneous maximization of $\bar{V}_{1:T,1}, \ldots, \bar{V}_{1:T,K}$. The Pareto optimality still holds when we replace the inequality $w_k \geq 1$ with $w_k \geq \rho_k^{\text{UB}}$, for any $\rho_k^{\text{UB}}$ that bounds the average $\bar{V}_{1:T,k}$ for any policy from above.

*State Space Exploration.* Consider visiting each state $s$ with empirical frequency as close as possible to a target frequency $\varrho_s$ in $T$ time steps, where $\varrho = \{\varrho_s\}_{s \in \mathcal{S}} \in \Delta^{\mathcal{S}}$. The task can be phrased as a CO-OMDP problem. For each state-action pair $(s, a)$, we define $V(s, a) \in \{0, 1\}^{\mathcal{S}}$ as the standard basis vector for $s$ in $\mathbb{R}^{\mathcal{S}}$, with value 1 at the $s$-coordinate and value 0 at the others. In addition, set $L_0 = 1, L_1 = \ldots = L_K = 0, U = \{\varrho\}$. Maximizing $g(\bar{V}_{1:T})$ is equivalent to minimizing the mean squared error $\sum_{s \in \mathcal{S}} (\varrho_s - \sum_{t=1}^{T} 1_{s_t=s}/T)^2$. To generalize, we can consider visiting certain subsets (not necessarily disjoint or covering) of $\mathcal{S}$ with some target frequencies.

Finally, when we specialize the CO-OMDP problem with $L_0 = 0$, we recover the SO-OMDP problem [28]. If we specialize with $\mathcal{S} = \{s\}$, we recover the BwR problem [2] with reward function $g$.

**Reachability of $\mathcal{M}$.** To learn the latent model, the agent has to travel among states. For any $s, s' \in \mathcal{S}$ and any stationary policy $\pi$, we define the travel time from $s$ to $s'$ under $\pi$ as the random variable $\Lambda(s'|\pi, s) := \min\{t : s_{t+1} = s', s_1 = s, s_{\tau+1} \sim p(\cdot|s_\tau, \pi(s_\tau)) \, \forall \tau\}$. We assume the following:

**Assumption 2.1.** *The latent CO-OMDP instance $\mathcal{M}$ is communicating, that is, the quantity $D := \max_{s,s' \in \mathcal{S}} \min_{\text{stationary } \pi} \mathbb{E}[\Lambda(s'|\pi, s)]$ is finite. We call $D$ the diameter of $\mathcal{M}$.*

The same reachability assumption is made in [28]. Since the instance $\mathcal{M}$ is latent, the corresponding diameter $D$ is also not known to the agent. Assumption 2.1 is weaker than the *unichain* assumption [6, 32, 41], where every stationary policy induces a single recurrent class on $\mathcal{S}$.

**Offline Benchmark and Regret.** To measure the effectiveness of a policy, we rephrase the agent's objective as the minimization of regret: $\text{Reg}(T) := \text{opt}(\mathsf{P}_{\mathcal{M}}) - g(\bar{V}_{1:T})$. The offline benchmark $\text{opt}(\mathsf{P}_{\mathcal{M}})$ is the optimum of the convex optimization problem $(\mathsf{P}(g))$, which serves as a fluid relaxation [38, 6] to the CO-OMDP problem.

$$(\mathsf{P}_{\mathcal{M}}): \quad \max_{x} \; g\left(\sum_{s \in \mathcal{S}, a \in \mathcal{A}_s} v(s, a) x(s, a)\right)$$

$$\text{s.t.} \sum_{a \in \mathcal{A}_s} x(s, a) = \sum_{s' \in \mathcal{S}, a' \in \mathcal{A}_{s'}} p(s|s', a') x(s', a') \quad \forall s \in \mathcal{S} \tag{2a}$$

$$\sum_{s \in \mathcal{S}, a \in \mathcal{A}_s} x(s, a) = 1 \tag{2b}$$

$$x(s, a) \geq 0 \qquad\qquad\qquad \forall s \in \mathcal{S}, a \in \mathcal{A}_s \tag{2c}$$

In ($\mathsf{P}_\mathcal{M}$), the variables $\{x(s,a)\}_{s,a}$ form a probability distribution over the state-action pairs. The set of constraints (2a) requires the rates of transiting into and out of each state $s$ to be equal.

We aim to design a non-anticipatory policy with an *anytime regret bound* $\text{Reg}(T) = O(1/T^\alpha)$ for some $\alpha > 0$. That is, for all $\delta > 0$, there exist constants $c, C$ (which only depend on $K, S, A, g, \delta$), so that the policy satisfies $\text{Reg}(T) \leq c/T^\alpha$ for all $T \geq C$ with probability at least $1 - \delta$. Achieving $\text{Reg}(T) = O(1/T^\alpha)$ for some $\alpha > 0$ implies achieving near-optimality, since $\text{opt}(\mathsf{P}_\mathcal{M})$ differs from the expected optimum only by an additive error of $O(\bar{L}D/T)$, by a similar reasoning to [28] (see [18] for details).

## 3 Challenges of CO-OMDP, and Algorithm TFW-UCRL2

We first discuss some unique challenges in the CO-OMDP, then present and discuss TFW-UCRL2 in Algorithm 1. Finally, we present the regret bound for TFW-UCRL2.

**Challenges.** We begin by describing some unique challenges in CO-OMDP hinted in the Introduction. Consider the three instances in Fig 1. An arc from state $s$ to $s'$ represents action $a$ with $p(s'|s,a) = 1$, and is labelled with its outcome $V(s,a)$, which is deterministic. Let's focus on Figs 1a, 1b. The common objective requires balancing the 2-dimensional outcomes by visiting the left loop (ll) and the right loop (rl) with frequency 0.5 each. In Fig 1a, the agent incurs a $O(1/T)$ regret by choosing ll once, then rl once, then ll once, and so on.

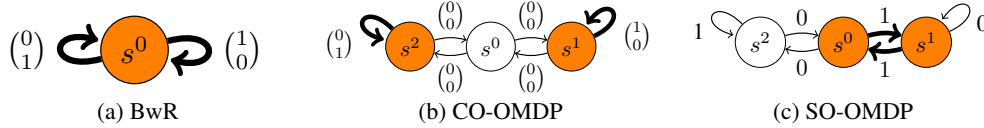

(a) BwR       (b) CO-OMDP       (c) SO-OMDP

Figure 1: Instances, with opt. actions bolded. Insts (1a, 1b) have $g(w) = -\sum_{k=1}^{2}(w_k - 0.5)^2/2$.

However, if the agent visits ll once, then rl once, then ll once, and so on in Fig 1b, she suffers $\text{Reg}(T) = \Omega(1)$. Indeed, she spends two third of the time at the actions with the 'sub-optimal' state $s^0$, resulting in $\bar{V}_{1:T} \approx (1/6, 1/6)^\top$ for large $T$. While the agent should visit each loop multiple times before going to state $s^0$ and then another loop, the length of stay at each loop is not *a priori* clear. Our Gradient Threshold Procedure (GTP) provides a principled way for determining these lengths, and GTP generalizes to other communicating MDPs. Finally, such a subtlety in state transitions does not occur in Fig 1c or generally in communicating SO-OMDP instances, where the agent achieves near-optimality by remaining in a single recurrent class.

**TFW-UCRL2** runs in episodes. Episode $m$ starts at the beginning of time $\tau(m)$ and ends at the end of time $\tau(m+1) - 1$. During episode $m$, the agent follows a certain stationary policy $\tilde{\pi}_m$. The start times $\{\tau(m)\}_{m=1}^\infty$ and policies $\{\tilde{\pi}_m\}_{m=1}^\infty$ are decided adaptively. We maintain confidence regions $H_m^v = \{H_m^v(s,a)\}_{s,a}$, $H_m^p = \{H_m^p(s,a)\}_{s,a}$ on the latent $v, p$ across episodes, by first defining

$$N_m(s,a) = \sum_{t=1}^{\tau(m)-1} 1_{(s_t,a_t)=(s,a)}, \quad N_m^+(s,a) = \max\{1, N_m(s,a)\}. \tag{3}$$

The estimates and confidence regions for $v$ are:

$$\hat{v}_m(s,a) := \frac{1}{N_m^+(s,a)} \sum_{t=1}^{\tau(m)-1} V_t(s_t,a_t) 1_{(s_t,a_t)=(s,a)}, \quad \text{rad}_{m,k}^v(s,a) = \tilde{O}\left(\sqrt{\frac{\hat{v}_{m,k}(s,a)}{N_m^+(s,a)}}\right),$$

$$H_m^v(s,a) := \left\{ \bar{v} \in [0,1]^K : |\bar{v}_k - \hat{v}_{m,k}(s,a)| \leq \text{rad}_{m,k}^v(s,a) \ \forall k \in [K] \right\}. \tag{4}$$

The estimates and confidence regions for $p$ are:

$$\hat{p}_m(s'|s,a) := \frac{1}{N_m^+(s,a)} \sum_{t=1}^{\tau(m)-1} 1_{(s_t,a_t,s_{t+1})=(s,a,s')}, \quad \text{rad}_m^p(s'|s,a) = \tilde{O}\left(\sqrt{\frac{\hat{p}_m(s'|s,a)}{N_m^+(s,a)}}\right),$$

$$H_m^p(s,a) := \left\{ \bar{p} \in \Delta^\mathcal{S} : |\bar{p}(s') - \hat{p}_m(s'|s,a)| \leq \text{rad}_m^p(s'|s,a) \ \forall s' \in \mathcal{S} \right\}. \tag{5}$$

We provide the complete expressions of $\mathsf{rad}_{m,k}^v(s,a), \mathsf{rad}_m^p(s'|s,a)$ in Appendix B.2. We now explain the three vital components of TFW-UCRL2: (i) Frank-Wolfe (FW) [21], which has been adapted in related research on BwR [2, 14] and exploration problems in MDPs [25, 41], (ii) Extended Value Iteration (EVI) [28], (iii) our crucial and novel Gradient Threshold Procedure (GTP).

**Frank Wolfe (FW) [21]** provides a way to balance the vectorial outcome at each time step $t$. We denote $\|\cdot\|_2$ as the Euclidean norm, and define $\Pi_U(w) = \mathrm{argmin}_{u \in U} \|u - w\|_2$. At time $t$, FW scalarizes the outcome in eqn (6) with the gradient

$$\nabla g(\bar{V}_{1:t-1}) = \frac{1}{K} \left[ (L_1, \ldots, L_K)^\top - L_0(\bar{V}_{1:t-1} - \Pi_U(\bar{V}_{1:t-1})) \right].$$

To gain intuitions, consider *State Space Exploration* with target frequency $\varrho$, where $L_0 = 1, L_1, \ldots, L_K = 0, U = \{\varrho\}$. The $s$-component of $\nabla g(\bar{V}_{1:t-1})$ is $(\varrho_s - \sum_{q=1}^{t-1} \mathbf{1}_{s_q=s}/(t-1))/K$, which encourages visiting state $s$ when its empirical frequency is below the target $\varrho_s$. Similarly, for *Multi-Objective Optimization* with KPI target $\rho$, the $k$-component of $\nabla g(\bar{V}_{1:t-1})$ is $(L_k + L_0 \max\{\rho_k - \bar{V}_{1:t-1,k}, 0\})/K$. The agent is motivated to focus on the $k$th objective when $\bar{V}_{1:t-1,k} \leq \rho_k$.

**Extended Value Iteration (EVI) [28]** solves for an optimistic stationary policy for an SO-OMDP problem, when $v, p$ are not known. We extract EVI from [28] in Appendix B.3. Ideally, at the start of each episode $m$, the agent wishes to compute the optimal policy under the scalarized reward $\nabla g(\bar{V}_{1:\tau(m)-1})^\top v$ and transition kernel $p$. Since $v, p$ are uncertain, the agent uses EVI [28] to compute the stationary policy $\tilde{\pi}_m$ in (7), which is optimal for the *optimistic choices* of $\tilde{v}_m \in H_m^v$ and $\tilde{p}_m \in H_m^p$. By optimistic choices $\tilde{v}_m, \tilde{p}_m$, we mean that the resulting single objective MDP with scalar rewards $\tilde{r}_m = \{\tilde{r}_m(s,a)\}_{s,a}, \tilde{r}_m(s,a) = \nabla g(\bar{V}_{1:\tau(m)-1})^\top \tilde{v}_m(s,a)$ and transition kernel $\tilde{p}$ has the highest long term average reward, among all $\bar{v}_m \in H_m^v, \bar{p}_m \in H_m^p$. The last argument $1/\sqrt{\tau(m)}$ of EVI is an additive error term allowed for EVI. By [28], EVI converges to a stationary policy $\tilde{\pi}_m$ in finite time when $H_m^p$ contains the transition kernel for a communicating MDP.

---

**Algorithm 1** TFW-UCRL2 on $g$

---

1: Inputs: Parameter $\delta \in (0,1)$, gradient threshold $Q \geq 0$ (default $Q = \bar{L}/\sqrt{K}$), initial state $s_1$.
2: Initialize $t = 1$
3: **for** Episode $m = 1, 2, \ldots$ **do**
4:     Set $\tau(m) = t$, and initialize $N_m^+(s,a)$ according to Eq (3) for each $s \in \mathcal{S}, a \in \mathcal{A}_s$.
5:     Compute the confidence regions $H_m^v, H_m^p$ respectively for $v, p$, according to Eqs (4, 5).
6:     Compute the optimistic reward $\tilde{r}_m = \{\tilde{r}_m(s,a)\}_{s \in \mathcal{S}, a \in \mathcal{A}_s}$:

$$\tilde{r}_m(s,a) = \max_{\bar{v}(s,a) \in H_m^v(s,a)} \nabla g(\bar{V}_{1:\tau(m)-1})^\top \bar{v}(s,a). \tag{6}$$

7:     Compute a $(1/\sqrt{\tau(m)})$-optimal optimistic policy $\tilde{\pi}_m$:

$$\tilde{\pi}_m, \leftarrow \mathsf{EVI}(\tilde{r}_m, H_m^p; 1/\sqrt{\tau(m)}). \tag{7}$$

8:     Initialize $\nu_m(s,a) = 0$ for each $s, a$, $\theta^{\mathsf{ref}} = \theta_{\tau(m)} = \nabla g(\bar{V}_{1:(\tau(m)-1)})$, $\Psi = 0$.
9:     **while** $\Psi \leq Q$ and $\nu_m(s_t, \tilde{\pi}_m(s_t)) < N_m^+(s_t, \tilde{\pi}_m(s_t))$ **do**
10:         Choose action $a_t = \tilde{\pi}_m(s_t)$.
11:         Observe the outcomes $V_t(s_t, a_t)$ and the next state $s_{t+1}$.
12:         Compute gradient $\theta_{t+1} = \nabla g(\bar{V}_t)$.           ▷ Frank-Wolfe
13:         Update $\Psi \leftarrow \Psi + \|\theta_{t+1} - \theta^{\mathsf{ref}}\|_2$.
14:         Update $\nu_m(s_t, a_t) \leftarrow \nu_m(s_t, a_t) + 1$.
15:         Update $t \leftarrow t + 1$.
16:     **end while**
17: **end for**

---

**The Gradient Threshold Procedure (GTP)** maintains FW's balancing effect on the vectorial outcomes, while overcoming the challenges in avoiding sub-optimal actions. GTP maintains a distance measure $\Psi$ on the gradients generated by FW during each episode, and starts the next episode if the measure $\Psi$ exceeds a threshold $Q$. A small $Q$ makes the agent alternate among different stationary

policies frequently and balances the outcomes, while a large $Q$ facilitates learning and avoids visiting sub-optimal states. A properly tuned $Q$ paths the way to solve the CO-OMDP problem.

A direct combination of FW and EVI corresponds to TFW-UCRL2 with $Q = 0$, which silences GTP and incurs $\text{Reg}(T) = \Omega(1)$ on the instance in Fig 1b. Let's assume start state to be $s^0$, the complete knowledge of $v, p$, and consistent tie breaking. The agent would go to $s^2$, take `ll` once, then go to $s^1$, take `rl` once, then back to $s^2$ and take `ll` once, and so on (the same dynamics as in **Challenges**). Indeed, under the pure effect of FW, the agent is obsessed with balancing the outcomes. Once $\bar{V}_{1:t-1,1} > \bar{V}_{1:t-1,2}$, the scalarized reward for $(s^2, \text{ll})$ is higher than that for $(s^1, \text{rl})$, and she travels to $s^2$. Similarly, once $\bar{V}_{1:t-1,1} \leq \bar{V}_{1:t-1,2}$, she travels to $s^1$. In this process, she is oblivious to the fact that constantly alternating between `ll`, `rl` penalizes her objective by constantly visiting $s^0$.

In contrast, applying TFW-UCRL2 with $0 < Q < \infty$ leads us to near-optimality. For example, with $Q = \bar{L}/\sqrt{K}$, the agent follows this interesting trajectory: Suppose the agent is at $s^0$ at time $t$. If $\bar{V}_{1:t-1,1} > \bar{V}_{1:t-1,2}$, then she would travel to $s^2$, take `ll` for $\Theta(\sqrt{Qt})$ times, then head back to $s^0$. Otherwise, she would travel to $s^1$, take `rl` for $\Theta(\sqrt{Qt})$ times, then head back to $s^0$. Altogether, for every $t$ we have $\bar{V}_{1:t-1,1}, \bar{V}_{1:t-1,2} = 0.5 \pm O(\sqrt{Q/t})$, and the agent only visits $s^0$ for $O(\sqrt{t/Q})$ times, leading to the anytime regret bound $\text{Reg}(T) = O((\sqrt{Q} + \sqrt{1/Q})/\sqrt{T})$.

Finally, in another extreme case of $Q = \infty$, in fact we have $\text{Reg}(T) = \Omega(1/SA \log T)$. Indeed, the condition $\Psi \leq Q$ is always satisfied. By applying [28], the agent alternates among `ll`, `rl` only $O(SA \log T)$ times in $T$ time steps. This leads to an imbalance in the outcomes, since the agent could stay at a loop for $\Omega(T/SA \log T)$ time, and results in $\text{Reg}(T) = \Omega(1/SA \log T)$.

**Main Results.** We establish regret bounds for TFW-UCRL2. Denote $S := |\mathcal{S}|$, $A := \frac{1}{S} \sum_{s \in \mathcal{S}} |\mathcal{A}_s|$, so $SA$ is the number of state-action pairs. Denote $\Gamma := \max_{s \in \mathcal{S}, a \in \mathcal{A}_s} \|p(\cdot|s,a)\|_0$, which is the maximum number of states from which a state-action pair can transit to. We employ the $\tilde{O}(\cdot)$ notation, which hides additive terms which scales with $\log(T/\delta)/T$ as well as multiplicative $\log(T/\delta)$ factors.

**Theorem 3.1.** *Consider TFW-UCRL2 with gradient threshold $Q > 0$, applied on a communicating CO-OMDP instance $\mathcal{M}$ with diameter $D$. With probability $1 - O(\delta)$, we have anytime regret bound*

$$\text{Reg}(T) = \tilde{O}\left(\left[\sqrt{L_0 Q} + \sqrt{L_0}\bar{L}D/\sqrt{KQ}\right] K^{1/4}\Big/\sqrt{T}\right) + \tilde{O}\left(\bar{L}(D+1)\sqrt{\Gamma SA}\Big/\sqrt{T}\right).$$

*In particular, setting $Q = \bar{L}/\sqrt{K}$ gives $\text{Reg}(T) = \tilde{O}(\bar{L}(D+1)\sqrt{\Gamma SA}/\sqrt{T})$.*

Let's focus on the first $\tilde{O}(\cdot)$ term in the bound. The $\tilde{O}(\sqrt{Q})$ term represents the regret due to the delay in gradient updates by GTP. The $\tilde{O}(1/\sqrt{Q})$ term represents the regret due to (a) the interference of GTP with the learning of $v, p$, (b) the switches among stationary policies, which could require visiting sub-optimal states. The second $\tilde{O}(\cdot)$ term is the regret due to the simultaneous exploration-exploitation by EVI.

By specializing $L_0 = 0, L_1, \ldots, L_K = 1$, TFW-UCRL2 incurs $\text{Reg}(T) = \tilde{O}(D\sqrt{\Gamma SA}/\sqrt{T})$ on SO-OMDP, which essentially matches [28][1]. By specializing $\mathcal{S} = \{s\}$, TFW-UCRL2 incurs $\text{Reg}(T) = \tilde{O}(\bar{L}\sqrt{A}/\sqrt{T})$, which matches [2] on BwR on $g$. While our regret bounds match [28, 2] in those special cases, the design and analysis of TFW-UCRL2 require novel ideas that depart from [28, 2]. We design the novel GTP for handling state transitions. In the upcoming analysis, we show that GTP is streamlined so that it achieves our regret bounds, without excessively interfering the balancing by FW and the learning by EVI. Finally, note that TFW-UCRL2 is a non-stationary policy that diversifies across different stationary policies across time. Interestingly, non-stationary policy is necessary achieving near-optimality, even when the model parameters are unchanging:

**Claim 3.2.** *Every stationary policy incurs an $\Omega(1)$ anytime regret on the instance in Fig 1b.*

The Claim, proved in Appendix B.4, illustrates a profound difference between communicating CO-OMDPs and unichain CO-OMDPs, see Appendix B.4.

**Max $\mathbb{E}[g(\bar{V}_{1:T})]$ vs max $g(\mathbb{E}[\bar{V}_{1:T}])$.** Our objective is to maximize $g(\bar{V}_{1:T})$ (also leads to max $\mathbb{E}[g(\bar{V}_{1:T})]$), which crucially different from maximizing $g(\mathbb{E}[\bar{V}_{1:T}])$. Now, for any policy, it holds that

$\mathbb{E}[g(\bar{V}_{1:T})] \le g(\mathbb{E}[\bar{V}_{1:T}]) \le \text{opt}(\mathsf{P}_{\mathcal{M}}) + O(\bar{L}D/T)$. The second inequality (formally proved in [18]) is demonstrated by showing that, for any policy, the empirical frequency of visiting each state-action pair is "nearly" a feasible solution to $(\mathsf{P}_{\mathcal{M}})$, with the $O(\bar{L}D/T)$ term capturing the error due to the near feasibility. Under TFW-UCRL2, we know that $\mathbb{E}[g(\bar{V}_{1:T})]$ tends to $\text{opt}(\mathsf{P}_{\mathcal{M}})$ as $T$ grows. Hence we also have $g(\mathbb{E}[\bar{V}_{1:T}])$ tending to $\text{opt}(\mathsf{P}_{\mathcal{M}})$ as $T$ grows.

Nevertheless, the converse is not true. Consider the instance in Fig 1b again, where the starting state is $s^0$. Consider the following policy: At the start, the agent transits to either $s^1$ or $s^2$ with probability $1/2$. After that, the agent loops at that state indefinitely. It is clear that $\Pr[\bar{V}_{1:T} = (1 - 1/T, 0)^\top] = \Pr[\bar{V}_{1:T} = (0, 1 - 1/T)^\top] = 1/2$. On the one hand, we have $g(\mathbb{E}[\bar{V}_{1:T}]) = -1/(8T^2)$, which tends to $\text{opt}(\mathsf{P}_{\mathcal{M}}) = 0$ as $T \to \infty$. On the other hand, we have $\mathbb{E}[g(\bar{V}_{1:T})] = -1/8 + O(1/T)$, which does not tend to $\text{opt}(\mathsf{P}_{\mathcal{M}}) = 0$ as $T \to \infty$. Altogether, solving $\max g(\mathbb{E}[\bar{V}_{1:T}])$ to near-optimality does not lead to the near-optimality for $\max \mathbb{E}[g(\bar{V}_{1:T})]$.

To this end, it is worth mentioning that the related works in the discounted settings [24, 6, 12, 43, 1, 25, 42, 29, 30, 33] focus on maximizing $\bar{g}(\mathbb{E}[\sum_{t=1}^\infty \gamma^t V_t(s_t, a_t)])$, where $\bar{g}$ is a certain non-linear function and $\gamma \in (0, 1)$ is the discounted factor. We envision that our technique could be useful for maximizing $\mathbb{E}[\bar{g}(\sum_{t=1}^\infty \gamma^t V_t(s_t, a_t))]$ instead of maximizing $\bar{g}(\mathbb{E}[\sum_{t=1}^\infty \gamma^t V_t(s_t, a_t)])$.

**Generalizations.** While Theorem 3.1 concerns the specialized aggregate function (1), Cheung [18] recently generalizes the algorithmic framework to any Lipschitz continuous and smooth function. By adapting to the online mirror descent algorithm [34], Cheung [18] proposes another algorithm that results in $O(1/T^3)$ regret (We hide the dependence on $\mathcal{M}$) for Lipschitz continuous concave aggregate functions that are not necessarily smooth.

## 4 Analysis of TFW-UCRL2

In this Section, we prove Theorem 3.1. To start, we consider events $\mathcal{E}^v$, $\mathcal{E}^p$ and Lemma 4.1, which is proved in Appendix C.1. The shorthand $\forall m, s, a$ means 'for all $m \in \mathbb{N}$, $s \in \mathcal{S}$, $a \in \mathcal{A}_s$'.

$$\mathcal{E}^v := \{v(s,a) \in H_m^v(s,a) \, \forall m, s, a\}, \quad \mathcal{E}^p := \{p(\cdot|s,a) \in H_m^p(s,a) \, \forall m, s, a\}.$$

**Lemma 4.1.** *It holds that* $\mathbb{P}[\mathcal{E}^v] \ge 1 - \delta/2, \mathbb{P}[\mathcal{E}^p] \ge 1 - \delta/2$.

We decompose $\text{Reg}(T)$ with the analytical tools on FW [21, 16], which is also adapted in [2, 14, 25, 41]. Define $v^* := \sum_{s,a} v(s,a)x^*(s,a)$, where $x^*$ is an optimal solution of $(\mathsf{P}_{\mathcal{M}})$. We have

$$g(\bar{V}_{1:t}) \ge g(\bar{V}_{1:t-1}) + \nabla g(\bar{V}_{1:t-1})^\top [\bar{V}_{1:t} - \bar{V}_{1:t-1}] - \frac{L_0}{K} \|\bar{V}_{1:t} - \bar{V}_{1:t-1}\|_2^2 \tag{8}$$

$$= g(\bar{V}_{1:t-1}) + \frac{1}{t}\nabla g(\bar{V}_{1:t-1})^\top [V_t(s_t, a_t) - \bar{V}_{1:t-1}] - \frac{L_0}{Kt^2}\|V_t(s_t, a_t) - \bar{V}_{1:t-1}\|_2^2$$

$$\ge g(\bar{V}_{1:t-1}) + \frac{1}{t}\nabla g(\bar{V}_{1:t-1})^\top [v^* - \bar{V}_{1:t-1}] + \frac{1}{t}\nabla g(\bar{V}_{1:t-1})^\top [V_t(s_t, a_t) - v^*] - \frac{L_0}{t^2}$$

$$\ge g(\bar{V}_{1:t-1}) + \frac{1}{t}\left[\text{opt}(\mathsf{P}_{\mathcal{M}}) - g(\bar{V}_{1:t-1})\right] + \frac{1}{t}\nabla g(\bar{V}_{1:t-1})^\top [V_t(s_t, a_t) - v^*] - \frac{L_0}{t^2}. \tag{9}$$

Step (8) is by the property that $g$ is $(2L_0/K)$-smooth w.r.t. $\|\cdot\|_2$ on the domain $[0,1]^K$ (see Appendix B.1). Rearranging (9) gives

$$t \cdot \text{Reg}(t) \le (t-1) \cdot \text{Reg}(t-1) + \frac{L_0}{t} + \nabla g(\bar{V}_{1:t-1})^\top [v^* - V_t(s_t, a_t)]. \tag{10}$$

Apply (10) recursively for $t = T, \dots, 1$, we obtain (recall that $\theta_t = \nabla g(\bar{V}_{1:t-1})$):

$$\text{Reg}(T) \le \frac{2L_0 \log T}{T} + \frac{1}{T}\sum_{t=1}^T \theta_t^\top [v^* - V_t(s_t, a_t)]. \tag{11}$$

The main analysis is now on the second term in (11), which requires novel technical analysis regarding the dynamics of the gradient threshold procedure. We start with the following bound.

**Proposition 4.2.** *Consider an execution of TFW-UCRL2 on a communicating instance with diameter $D$. For each $T \in \mathbb{N}$, suppose that there is a deterministic constant $M(T)$ s.t. $\Pr[m(T) \leq M(T)] = 1$. Conditioned on events $\mathcal{E}^v, \mathcal{E}^p$, with probability at least $1 - O(\delta)$ we have*

$$\sum_{t=1}^{T} \theta_t^\top [v^* - V_t(s_t, a_t)] = \tilde{O}\left((Q\sqrt{K} + \bar{L}D)M(T)\right) + \tilde{O}\left(\bar{L}(D+1)\sqrt{\Gamma SAT}\right).$$

Proposition 4.2 bounds two sources of error: (i) the error due to GTP, (ii) the estimation errors associated with $H_m^v, H_m^p$ and EVI. Error (ii) can be upper bounded by the machinery in [28]. Error (i) concerns the following discrepancy. For each time $t$ in episode $m$, the action $a_t = \tilde{\pi}_m(s_t)$ is chosen based on policy $\tilde{\pi}_m$, which involves the scalarization by $\theta_{\tau(m)}$. However, ideally the action at time $t$ should balance the current vectorial outcomes by the scalarization with $\theta_t$. The Proposition bounds error (i) by charging the discrepancy to the threshold $Q$ and the upper bound $M(T)$. To complete the proof of bounding $\text{Reg}(T)$, we establish a bound $M(T)$ small enough to achieve Theorem 3.1.

**Lemma 4.3.** *Consider an execution of TFW-UCRL2 with gradient threshold $Q > 0$. With certainty, for every $T \in \mathbb{N}$ we have $m(T) \leq M(T) = \tilde{O}(\sqrt{L_0 T/(\sqrt{K}Q)})$.*

The Lemma bounds the error of GTP in balancing the outcomes. Under FW, the gradients change at a rate $O(1/t)$, which is slow enough for the agent to judiciously delay the gradient updates without sacrificing their balancing effect too much. This opens the door to avoid visiting sub-optimal states too frequently.

*Sketch Proof of Lemma 4.3.* First, observe that $\{1, \ldots, m(T)\}$ is the union of

$$\mathcal{M}_\Psi(T) := \{m \in \mathbb{N} : \tau(m) \leq T, \text{ episode } m+1 \text{ is started due to } \Psi \geq Q\},$$
$$\mathcal{M}_\nu(T) := \{m \in \mathbb{N} : \tau(m) \leq T, \text{ episode } m+1 \text{ is started due to}$$
$$\nu_m(s_t, \tilde{\pi}_m(s_t)) \geq N_m^+(s_t, \tilde{\pi}_m(s_t)) \text{ for some } t \geq \tau(m)\},$$

To prove the Lemma, it suffices to show that:

$$|\mathcal{M}_\Psi(T)| \leq M_\Psi(T) := 1 + (\sqrt{K}Q/2L_0) + 4\sqrt{2L_0 T/(\sqrt{K}Q)}, \tag{12}$$
$$|\mathcal{M}_\nu(T)| \leq M_\nu(T) := SA(1 + \log_2 T). \tag{13}$$

The bound (13) follows from [28]. Thus, we focus on showing bound (12). Let's express $\mathcal{M}_\Psi(T) = \{m_1, m_2, \ldots, m_{n_\Psi}\}$, where $m_1 < m_2 < \ldots < m_{n_\Psi}$. We also define $m_0 = 0$. We focus on an episode index $m_j$ with $j \geq 1$, and consider for each $t \in \{\tau(m_j) + 1, \ldots, \tau(m_j + 1)\}$ the difference $\|\theta_t - \theta_{\tau(m_j)}\|_2$. In the following, we argue that the gradients under FW changes slowly:

$$\|\theta_t - \theta_{\tau(m_j)}\|_2 = \|\nabla g(V_{1:t-1}) - \nabla g(V_{1:\tau(m_j)-1})\|_2$$

$$\leq \frac{L_0}{K} \left\| \frac{1}{t-1} \sum_{q=1}^{t-1} V_q(s_q, a_q) - \frac{1}{\tau(m_j) - 1} \sum_{q=1}^{\tau(m_j)-1} V_q(s_q, a_q) \right\|_2$$

$$= \frac{L_0}{K} \cdot \frac{t - \tau(m_j)}{t-1} \cdot \left\| \frac{1}{t - \tau(m_j)} \sum_{q=\tau(m_j)}^{t-1} V_q(s_q, a_q) - \frac{1}{\tau(m_j) - 1} \sum_{q=1}^{\tau(m_j)-1} V_q(s_q, a_q) \right\|_2$$

$$\leq \frac{2L_0}{\sqrt{K}} \cdot \frac{t - \tau(m_j)}{t-1} \leq \frac{2L_0}{\sqrt{K}} \cdot \frac{t - \tau(m_j)}{\tau(m_j)}.$$

Since $m_j \in \mathcal{M}_\Psi(T)$, we know that $\sum_{t=\tau(m_j)}^{\tau(m_j+1)} \|\theta_t - \theta_{\tau(m_j)}\|_2 > Q$, which means

$$\frac{(\tau(m_j+1) - \tau(m_{j-1}+1))^2}{\tau(m_{j-1}+1)} \geq \frac{(\tau(m_j+1) - \tau(m_j))^2}{\tau(m_j)} \geq \sum_{t=\tau(m_j)}^{\tau(m_j+1)} \frac{t - \tau(m_j)}{\tau(m_j)} > \frac{\sqrt{K}Q}{2L_0}, \tag{14}$$

Inequality (14) says that, since gradients change slowly, the time indexes $\{\tau(m_j+1)\}_{j=1}^{n_\Psi}$ have to be far apart. Thus, $n_\Psi$ can be bounded from above. Indeed, by some technical arguments (see Appendix C.2), inequality (14) turns out to imply $\tau(m_{\lceil Q' \rceil + j} + 1) \geq Q'(j-1)^2/16$, where $Q' = \sqrt{K}Q/(2L_0)$. With $j = n_\Psi - 1$, we get $Q'(n_\Psi - 2)^2/16 \leq \tau(m_{\lceil Q' \rceil + n_\Psi - 1} + 1) \leq T$, leading to (12). $\square$

Combining the bound (11), Proposition 4.2 and Lemma 4.3, we have proved Theorem 3.1.

# 5 Numerical Experiments

We empirically evaluate TFW-UCRL2 on *State Space Exploration* on 3 instances: Small, Medium, Large. These instances are detailed in Appendix A.1. In Fig 2a, TFW-UCRL2 is simulated on each instance and each $Q$ for 25 times. In Figs 2b, 2c, TFW-UCRL2 is simulated on each instance and $Q = \bar{L}/\sqrt{K}$ for 25 times. Each curve plots the averages across the 25 trials, and each error bar quantifies the $\pm$ standard deviation error region. Fig 2a depicts $\text{Reg}(10^5)$ under different $Q$s. While extremely small or large $Q$ leads to a large regret, TFW-UCRL2 seems robust in the middle range of $Q$. Our default $Q = \bar{L}/\sqrt{K}$ (green dot) is motivated by our analysis, and it does not optimize the empirical performance. Tuning $Q$ online is an interesting research direction.

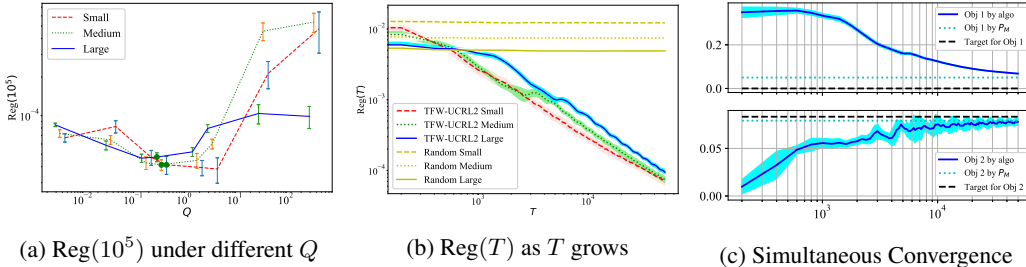

(a) $\text{Reg}(10^5)$ under different $Q$     (b) $\text{Reg}(T)$ as $T$ grows     (c) Simultaneous Convergence

Figure 2: Simulation Results on *State Space Exploration*

Fig 2b demonstrates the trend of $\text{Reg}(T)$ as $T$ grows, in log-log scales. The performance of TFW-UCRL2 is in contrast with the random policy (in yellow), which samples an action uniformly at random at every state. The $\text{Reg}(T)$ under TFW-UCRL2 converges to 0 as $T$ grows, while the $\text{Reg}(T)$ under the random policy is constant. The slight wiggling in the plots for TFW-UCRL2 is due to GTP, which could deteriorate the objective in the short term but still leads to near-optimality eventually.

Fig. 2c highlights the simultaneous convergence of each objective to its target on the Large instance. The instance involves a star graph with a center state and 12 branch states. The objectives are to visit the center state with frequency 0 (Obj 1) and to visit each branch state with frequency $1/12 = 0.83$ (Objs $2, \ldots 13$). These target frequencies (in dashed black) are not realizable, and we plot (in dotted cyan) the frequencies indicated by $v^* = \sum_{s,a} v(s,a)x^*(s,a)$, where $x^*$ is an optimal solution of $(\mathsf{P}_\mathcal{M})$. Along with the complete plot in Fig 4 in Appendix A.2, we see that the outputs $\{V_{1:T,k}\}_{k=1}^{13}$ by TFW-UCRL2 (in solid blue) *simultaneously converge* to all the 13 target frequencies.

## Footnotes

[1]Jaksch et al. [28] achieve the regret bound $\tilde{O}(DS\sqrt{A}/\sqrt{T})$. The factor of $S$ is improved to $\sqrt{\Gamma S}$, by applying an empirical Bernstein inequality[7] instead of the Hoeffding inequality, as used in [23].

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
