[Supplementary Material]

# Contents

# A  Numerical Experiments (Cont'd)

In this Appendix Section, we provide in Appendix Section A.1 the details about the set up for our numerical experiments. We describe the evaluated instances for the *State Space Exploration* and the *Multi-Objective Optimization* problems. Then, in Appendix Section A.2, we provide additional numerical results for the *State Space Exploration* problem studied in the main text, as well as the numerical results for the *Multi-Objective Optimization* problem.

## A.1  Setup Details

Our simulation study is conducted on star-shaped instances. We first describe a parameteric procedure GENSTAR$(S, c)$, which returns a tuple $(\mathcal{S}, s_1, \mathcal{A}, p)$ that constitutes an MDP on a star graph, given a positive integer $S$ and a non-negative integer $c$.

The procedure GENSTAR$(S, c)$ returns the following $(\mathcal{S}, s_1, \mathcal{A}, p)$:

- State space $\mathcal{S} = \{0, 1, \ldots S\}$. State 0 is the center state, while states $1, \ldots, S$ are the branch states.
  - From the center state, the agent can travel to any state within one state transition.
  - From a branch state, the agent can either stay at that branch state or travel to the center state within one state transition.
- The starting state $s_1$ is the center state 0.
- Action collection $\{\mathcal{A}_s\}_{s=0}^S$. The set $\mathcal{A}_0$ is for the center node 0, and for each $s \in \{1, \ldots S\}$ the set $\mathcal{A}_s$ is for the branch state $s$.
  - The action set $\mathcal{A}_0$ contains $S$ 'good' actions $a_1, \ldots, a_S$ and $c$ 'bad' actions $b_1, \ldots, b_c$.
  - The action set $\mathcal{A}_s$ contains 2 'good' actions $a_0, a_s$, and $c$ 'bad' actions $b_1, \ldots, b_c$.
- Transition kernel $p$:
  - For the center state 0:
    * For each $1 \leq s \leq S$, we have $p(s|0, a_s) = 0.95 = 1 - p(0|0, a_s)$,
    * For each $b \in \{b_1, \ldots b_c\}$, we have $p(s'|0, b) = 1/(S+1)$ for $s' \in \{0, \ldots, S\}$.
  - For each branch state s:
    * We have $p(s|s, a_s) = 0.95 = 1 - p(0|s, a_s)$, and $p(0|s, a_0) = 0.95 = 1 - p(s|s, a_0)$.
    * For each $b \in \{b_1, \ldots b_c\}$, we have $p(s'|s, b) = 1/(S+1)$ for $s' \in \{0, \ldots, S\}$.

**State Space Exploration.** We generate three instances: Small, Medium, Large. We define them with GENSTAR. To complete the description of these instances, we discuss on how the vectorial outcomes $\{V(s, a)\}_{s,a}$ and the target frequency vector $\varrho$ are defined. Recall that in the State Space Exploration problem, the vectorial outcome $V(s, a)$ is (with certainty) equal to the $s$th standard basis vector $e_s$ in $\mathbb{R}^S$, where $e_s$ has value 1 at the $s$-coordinate and 0 at the other. For the tuple returned by GENSTAR$(S, c)$, we define the target frequency $\varrho$ as follows. For the center state 0, we define $\varrho = 0$. For each branch state $s \in \{1, \ldots, S\}$, we define $\varrho_s = 1/S$.

With the discussions above, we see that once we specify the tuple $(\mathcal{S}, s_1, \mathcal{A}, p)$ by GENSTAR, the whole problem instance for the State Space Exploration problem is fully defined. The instances Small, Medium, Large are generated as follows:

- Small: GENSTAR$(5, 2)$,
- Medium: GENSTAR$(8, 3)$,
- Large: GENSTAR$(12, 4)$.

**Multi-Objective Optimization.** We generate three instances: Small, Medium, Large. We define them with GENSTAR. To complete the description of these instances, we discuss on how the vectorial outcomes $\{V(s, a)\}_{s,a}$, the KPI target vector $\rho$, and $L_0, L_1, \ldots L_K$ are defined.

To define $\{V(s, a)\}_{s,a}$, we first need to specify $K$, the dimension of each $V(s, a)$. For the center state 0, we define $v(s)$ to be the $K$-dimensional zero vector. For each branch state $s \in \{1, \ldots, S\}$,

we generate a random vector $\upsilon(s) \in \mathbb{R}^K$, where $\lceil K/2 \rceil$ randomly chosen coordinates in $\upsilon(s)$ are uniformly distributed across $[0.3, 1]$ and the other coordinates are 0. Lastly, for each state-action pair $s, a$ and each $k \in \{1, \ldots, K\}$, we define $V_k(s, a)$ to be the Bernoulli random variable with mean $\upsilon_k(s)$. Next, the KPI target $\rho \in \mathbb{R}^K$ is defined as $\rho = \sum_{s \in \mathcal{S}} w(s)\upsilon(s)$, where $w \in \Delta^{\mathcal{S}}$ is a randomly chosen probability distribution on $\mathcal{S}$. Finally, we specify $L_0 = L_1 = \ldots = L_K = 1$.

With the discussions above, we see that once we specify the tuple $(\mathcal{S}, s_1, \mathcal{A}, p)$ by GENSTAR and $K$, the whole problem instance for *Multi-Objective Optimization* is fully defined. The instances Small, Medium, Large are generated as follows:

- Small: GENSTAR$(5, 2)$, $K = 3$,
- Medium: GENSTAR$(8, 3)$, $K = 5$,
- Large: GENSTAR$(12, 4)$, $K = 6$.

## A.2 Additional Numerical Results for State Space Exploration

Figure 3: Regret for State Space Exploration Instances (large version of Fig 2b)

First, we provide a larger plot in Fig. 3 for Fig 2b. With a closer view, we have the following two observations. First, while the random policy incurs $\text{Reg}(T) = \Omega(1)$ in all the three instances, we see that the regret decreases when the instance becomes large. Indeed, a larger state space facilitates uniform exploration when actions are chosen uniformly at random. Second, for the plots on the performance of TFW-UCRL2, note that the error bars in the regret curve increase slightly when the problem size decrease. The reason should similar to the reason of the first observation. With a smaller state space, the deviation from the target frequency at each branch state has a greater impact on the regret bound. Hence, the variation in the frequency of visiting a branch state would affect the regret to a higher degree when the number of branches becomes smaller.

Next, we demonstrate in Fig 4 the full picture for the simultaneous convergence of each state's visit frequency to its target frequency in the Large instance. By simultaneous convergence, we mean that if we perform *one trial* of TFW-UCRL2 for sufficiently many time steps $T$, then we observe that $\sum_{t=1}^T \mathbf{1}(s_t = s)/T \to \varrho_s$ simultaneously for all $s \in \mathcal{S}$ (recall that $\mathcal{S} = \{0, 1, \ldots 12\}$, and $\varrho = (0, 1/12, 1/12, ..., 1/12)$). Since the error bar diminishes in each of the 13 plots, we conclude that TFW-UCRL2 succeeds in the desired simultaneous convergence as time progresses. Interestingly, the trend of convergence for each state is distinct from one another, even when the instance is

Figure 4: *State Space Exploration* on the Large instance

symmetric. In comparison to Fig 3, which demonstrate the error bars for standard deviation of the average error from $\varrho$, Fig 4 has larger error bars.

## A.3    Numerical Results for Multi-Objective Optimization

The plots in Figs 5, 6 display our simulation results on the three *Multi-Objective Optimization* instances: Small, Medium, Large. Figs 5, 6 are respectively the *Multi-Objective Optimization* versions of Figs 3, 4.

Figure 5: Regret for *Multi-Objective Optimization*

Figure 6: *Multi-Objective Optimization* on the Large instance

In Fig 5, we observe that TFW-UCRL2 converges much more slowly in the case of *Multi-Objective Optimization* than the case of *State Space Exploration* (see Fig 3). Indeed, for *Multi-Objective Optimization*, both $v, p$ are not known, while for *State Space Exploration* only $p$ is not known. Apart from the difference in speed of convergence, Figs 3, 5 show similar trends in the performance of the random policy and our TFW-UCRL2.

Fig 6 showcases the simultaneously convergence of the TFW-UCRL2 to each of the 6 KPI targets as time progresses. Recall that in the Target Set Objective problem, it suffices to have TFW-UCRL2's performance $\bar{V}_{1:T,k}$ (solid blue line) to be above the KPI target $\rho_k$ (dash black line), for each $1 \leq k \leq 6$. However, in this instance (Large instance) the KPI targets are not realizable. Therefore in most of the plots in Fig 6, the algorithm approaches the KPI targets from below. Overall, we observe a much slower convergence here than the previous case of State Space Exploration. In addition, the TFW-UCRL2's performance $\bar{V}_{1:T}$ (solid blue lines) appears to be quite different from the offline benchmark solution $\sum_{s,a} v(s,a)x^*(s,a)$ (dotted cyan line).

# B Supplementary Details on CO-OMDP and TFW-UCRL2

## B.1 Properties of the Reward Function $g$

In this Appendix Section, we demonstrate the concavity, the Lipschitz continuity and smoothness of the reward function $g$. Now, recall the projection function $\Pi_U(w) = \text{argmin}_{u \in U} \|w - u\|_2$. By the compactness of $U$, the argmin $\Pi_U(w)$ exists, and it is unique by the convexity of $U$.

**Concavity.** It suffices to show that the function $f(w) := \min_{u \in U}\{\|w - u\|_2^2\}$ is convex in $w$. For any $w, w' \in [0,1]^K$, we readily see that

$$
\begin{aligned}
f\left(\frac{w + w'}{2}\right) &= \min_{u \in U}\left\{\left\|\frac{w + w'}{2} - u\right\|_2^2\right\} \\
&\leq \left\|\frac{w + w'}{2} - \frac{\Pi_U(w) + \Pi_U(w')}{2}\right\|_2^2 \qquad (15) \\
&\leq \left[\frac{1}{2}\|w - \Pi_U(w)\|_2 + \frac{1}{2}\|w' - \Pi_U(w')\|_2\right]^2 \\
&= \frac{1}{4}\|w - \Pi_U(w)\|_2^2 + \frac{1}{4}\|w' - \Pi_U(w')\|_2^2 + \frac{1}{2}\|w - \Pi_U(w)\|_2\|w' - \Pi_U(w')\|_2 \\
&\leq \frac{1}{2}\|w - \Pi_U(w)\|_2^2 + \frac{1}{2}\|w' - \Pi_U(w')\|_2^2 = \frac{1}{2}\left(f(w) + f(w')\right). \qquad (16)
\end{aligned}
$$

Step (15) is by the convexity of $U$, which ensures that $(\Pi_U(w) + \Pi_U(w'))/2 \in U$. Step (16) is by the Cauchy-Schwartz inequality.

**Lipschitz Continuity.** We next demonstrate that $g$ is $(\bar{L}/\sqrt{K})$-Lipschitz continuous w.r.t. $\|\cdot\|_2$ on the domain $[0,1]^K$, where $\bar{L} = L_0 + \max_{1 \leq k \leq K}|L_K|$. The Lipschitz continuity is an immediate consequence of expressing the gradient of $g$:

$$
\nabla g(w) = \frac{1}{K}\left[(L_1, \ldots, L_K)^\top - L_0 \cdot (w - \Pi_U(w))\right].
$$

Clearly, we have $\|\nabla g(w)\|_2 \leq \bar{L}/\sqrt{K}$, hence establishing the required Lipschitz continuity.

**Smoothness.** Finally, we argue that $g$ is $(2L_0/K)$-smooth w.r.t. the norm $\|\cdot\|_2$ over the domain $[0,1]^K$, that is, for any $w, w' \in [0,1]^K$ we have

$$
\|\nabla g(w) - \nabla g(w')\|_2 \leq \frac{L_0}{K}\left[\|w - w'\|_2 + \|\Pi_U(w) - \Pi_U(w')\|_2\right] \leq \frac{2L_0}{K}\|w - w'\|_2.
$$

## B.2 Full Expressions of $\text{rad}_{m,k}^v(s,a)$, $\text{rad}_m^p(s'|s,a)$

Define $(\text{log-}v)_m := \log(12KSA\tau^2(m)/\delta)$. The full expression of $\text{rad}_{m,k}^v(s,a)$ is

$$
\text{rad}_{m,k}^v(s,a) := \sqrt{\frac{2\hat{v}_{m,k}(s,a) \cdot (\text{log-}v)_m}{N_m^+(s,a)}} + \frac{3 \cdot (\text{log-}v)_m}{N_m^+(s,a)}.
$$

Define $(\text{log-}p)_m := \log(12S^2A\tau^2(m)/\delta)$. The full expression of $\text{rad}_m^p(s'|s,a)$ is

$$\text{rad}_m^p(s'|s,a) := \sqrt{\frac{2\hat{p}_m(s'|s,a)\cdot(\text{log-}p)_m}{N_m^+(s,a)}} + \frac{3\cdot(\text{log-}p)_m}{N_m^+(s,a)}.$$

## B.3 Extended Value Iteration [28]

We present the Extended Value Iteration (EVI) algorithm from [28] in Algorithm 2. When we apply the EVI algorithm at the start of the $m$th episode, the input $\tilde{r}$ is an optimistic estimate of the scalarized reward, where $\tilde{r}(s,a) = \max_{\bar{v}(s,a)\in H_m^v(s,a)} \nabla g(\bar{V}_{1:\tau(m)-1}(s,a))^\top \bar{v}(s,a)$. The input $H^p$ is a confidence region that contains the latent transition kernel $p$ with high probability, and $\epsilon \in (0,1)$ is an error parameter.

---

**Algorithm 2** $\text{EVI}(\tilde{r}, H^p; \epsilon)$, mostly extracted from [28]

1: Initialize VI record $u_0 \in \mathbb{R}^{\mathcal{S}}$ as $u_0(s) = 0$ for all $s \in \mathcal{S}$.
2: **for** $i = 0,1,\ldots$ **do**
3:    For each $s \in \mathcal{S}$, compute VI record

$$u_{i+1}(s) = \max_{a\in\mathcal{A}_s} \tilde{\Upsilon}_i(s,a), \text{ where } \tilde{\Upsilon}_i(s,a) = \tilde{r}(s,a) + \max_{\bar{p}\in H^p(s,a)} \left\{ \sum_{s'\in\mathcal{S}} u_i(s')\bar{p}(s') \right\}.$$

4:    **if** $\max_{s\in\mathcal{S}}\{u_{i+1}(s) - u_i(s)\} - \min_{s\in\mathcal{S}}\{u_{i+1}(s) - u_i(s)\} \le \epsilon$ **then**
5:        Break the **for** loop.
6:    **end if**
7: **end for**
8: Define stationary policy $\tilde{\pi} : \mathcal{S} \to \mathcal{A}_s$ as $\tilde{\pi}(s) = \text{argmax}_{a\in\mathcal{A}_s} \tilde{\Upsilon}_i(s,a)$.
9: Define an optimistic dual solution $\tilde{\phi} = \max_{s\in\mathcal{S}}\{u_{i+1}(s) - u_i(s)\}$, $\tilde{\gamma} = u_i$.
10: Return policy $\tilde{\pi}$.
11: Auxiliary output: dual variables $(\tilde{\phi}, \tilde{\gamma})$.

---

## B.4 Proof of Claim 3.2

**Claim 3.2.** *Every stationary policy incurs an $\Omega(1)$ anytime regret on the instance in Fig 1b.*

**Remarks.** Before proving the Claim, we comment on our problem of communicating CO-OMDPs as compared to the problems of BwR [2, 3, 14, 17] and unichain MDPs [32, 36, 41]. It is well established that one can achieve $\text{Reg} = o(T)$ with suitable stationary (possibly randomized) policies, for BwR [2] and for single [38] or multi-objective MDPs [6]. By Claim 3.2, our communicating CO-OMDP problem is markedly different from those two cases, and requires a different approach from those related works.

To elaborate on the unichain case, it is shown that for a unichain multi-objective MDP instance $\mathcal{M}$, the optimal solution $x^*$ of the corresponding offline benchmark $(\mathsf{P}_{\mathcal{M}})$ satisfies $\sum_{a\in\mathcal{A}_s} x^*(s,a) > 0$ for all $s \in \mathcal{S}$. By following the stationary policy that chooses action $a \in \mathcal{A}_s$ at state $s$ with probability $x^*(s,a)/\sum_{a'\in\mathcal{A}_s} x^*(s,a')$, it can be shown that $\lim_{T\to\infty} g(\frac{1}{T}\sum_{t=1}^T V_t(s_t,a_t)) = \text{opt}(\mathsf{P}_{\mathcal{M}})$ with the technique in [6]. With mixing time assumptions along the line of [36, 41], the previous convergence in limit can be made non-asymptotic in a form of a regret bound, but the regret bound would depend on certain mixing time parameters, similar to [36, 41].

In contrast, with the communicating MDP assumption, we cannot hope to achieve $\text{Reg}(T) = o(T)$ by a similar approach. Indeed, solving $(\mathsf{P}_{\mathcal{M}})$ for the instance in Fig 1b gives the optimal solution $x^*(s^1,\texttt{ll}) = x^*(s^2,\texttt{rl}) = 1/2$, and $x^*(s,a) = 0$ for all other state action pairs. We cannot construct a stationary policy in the same way as the unichain case, since $\sum_{a\in A_{s^0}} x^*(a,s^0) = 0$. In fact, a more subtle issue is that the optimal solution $x^*$ does not inform the agent on how to act when she is at state $s^0$. Indeed, as demonstrated in Claim 3.2 and also in the discussion about our Gradient Threshold Procedure (GTP) in Section 3, the choice of action at $s^0$ very much depend on the numbers

of visits to `ll`, `rl` in the previous rounds, which leads to a non-stationary policy that is not implied by the solution $x^*$.

Finally, apart from overcoming the difficulty in working with $x^*$, our method also bypasses the mixing time assumptions in [36, 41], which requires that all deterministic stationary policies jointly satisfy some mixing time bounds. Our method (which harnesses [28] but also crucially depends on our GTP) only needs to assume bounded diameters, which is much less severe than the assumptions of mixing time [28].

*Proof of Claim 3.2.* Denote the state space of the instance in Fig 1b as $\mathcal{S}_{1b} = \{s^0, s^1, s^2\}$, and recall that the objective function is $g(w) = -\sum_{k=1}^{2}(w_k - 0.5)^2/2$. A stationary policy, be it deterministic or randomized, induces a time homogeneous Markov chain $(S_{1b}, \mathfrak{p})$ on the instance in Fig 1b, where $\mathfrak{p}(s'|s)$ is the probability of transiting from $s \in S_{1b}$ to $s' \in S_{1b}$ in the Markov chain. Every state transition occurs along an arc in the instance in Fig 1b.

We prove the claim by inspecting $(S_{1b}, \mathfrak{p})$ under different cases on $\mathfrak{p}$. If $\mathfrak{p}(s^k|s^k) = 0$ for some $k \in \{1, 2\}$, then clearly $V_{1:t,k} = 0$ for every $t$, leading to $\text{Reg}(T) = \Omega(1)$ for all $T$. Else, suppose that $\mathfrak{p}(s^k|s^0) = 0$ for some $k \in \{1, 2\}$. This means that the agent cannot reach $s^k$ from $s^0$, which still leads to $\text{Reg}(T) = \Omega(1)$ for all $T$. Else, suppose that $\mathfrak{p}(s^0|s^k) = 0$ for some $k \in \{0, 1\}$. This means that the agent cannot leave the state $s^k$ once she reaches $s^k$. Therefore, under the stationary policy, she either never visits $s^k$, or she does visit $s^k$, but she will not be able to visit $s^{3-k}$ forever. This means that $\text{Reg}(T) = \Omega(1)$ for sufficiently large $T$.

The remaining case is when $\mathfrak{p}(s|s') > 0$ for all arcs from $s$ to $s'$ in the instance. In this case, all states in $S_{1b}$ forms a single recurrent class. By either the Perron-Frobenius Theorem or Theorem 1.7.5 in [35], the stationary distribution $\{\lambda_s\}_{s \in S_{1b}}$ is entry-wise positive, and in particular $\lambda_{s^0} > 0$. This implies that $\lim_{T \to \infty} \mathbb{E}[\sum_{t=1}^{T} \mathbf{1}(a_t \in \mathcal{A}_{s^0})]/T > 0$, which further implies that $\lim_{T \to \infty} \mathbb{E}[|\frac{1}{T}\sum_{t=1}^{T} \mathbf{1}(a_t = \text{ll}) - \frac{1}{2}| + |\frac{1}{T}\sum_{t=1}^{T} \mathbf{1}(a_t = \text{rl}) - \frac{1}{2}|] < 1$. The mentioned time averages exist since the Markov chain $(S_{1b}, \mathfrak{p})$ is recurrent and aperiodic. As a result, we have $\lim_{T \to \infty} g(\bar{V}_{1:T}) < 0 = \text{opt}(\mathsf{P}_{\mathcal{M}})$, which means that $\text{Reg}(T) = \Omega(1)$ for sufficiently large $T$. $\qquad\square$

## C  Supplementary Proofs for Proving Theorem 3.1

### C.1  Proof of Lemma 4.1 for events $\mathcal{E}^v, \mathcal{E}^p$

**Lemma 4.1.** *It holds that $\mathbb{P}[\mathcal{E}^v] \geq 1 - \delta/2, \mathbb{P}[\mathcal{E}^p] \geq 1 - \delta/2$.*

The proof of the Lemma uses the following Theorem by [7].

**Theorem C.1** ([7]). *Let random variables $Y_1, \ldots, Y_N \in [0,1]$ be independently and identically distributed. Consider their sample mean $\hat{Y}_N$ and their sample variance $\hat{\sigma}_{Y,N}^2$:*

$$\hat{Y}_N = \frac{1}{N}\sum_{i=1}^{N} Y_i, \quad \hat{\sigma}_{Y,N}^2 = \frac{1}{N}\sum_{i=1}^{N}(Y_i - \hat{Y})^2.$$

*For any $\delta \in (0,1)$, the following inequality holds:*

$$\Pr\left(\left|\hat{Y}_N - \mathbb{E}[Y_1]\right| \leq \sqrt{\frac{2\hat{\sigma}_{Y,N}^2 \log(1/\delta)}{N}} + \frac{3\log(1/\delta)}{N}\right) \geq 1 - 3\delta. \qquad\blacksquare$$

*Proof of Lemma 4.1.* We first analyze event $\mathcal{E}^v$. Consider a fixed objective index $k$, a fixed state $s$ and a fixed action $a$. We assert that

$$\mathbb{P}\left[|\hat{v}_{m,k}(s,a) - v_k(s,a)| \leq \text{rad}_{m,k}^v(s,a) \text{ for all } m\right] \geq 1 - \frac{\delta}{2KSA}. \tag{17}$$

Assuming inequality (17), the bound $\Pr[\mathcal{E}^v] \geq 1 - \delta/2$ is established by taking a union bound over $s \in \mathcal{S}$, $a \in \mathcal{A}_s$ and $k \in [K]$.

We establish inequality (17) by applying Theorem C.1 and the union bound. First, note that $\hat{v}_{m,k}(s,a)$ is the sample mean of $N_m(s,a)$ i.i.d. random variables, which are distributed as $V_k(s,a)$. Let $\hat{\sigma}^2_{v,m,k}$ be the sample variance of these $N_m(s,a)$ i.i.d random variables. To apply the union bounds, we also consider $\Upsilon^V_1, \ldots, \Upsilon^V_T$, which are $T$ i.i.d samples with the same distribution as $V_k(s,a)$. Denote $\hat{\Upsilon}^V_t$, $\hat{\sigma}^2_{\Upsilon^V,t}$ respectively as the sample mean and variance of $\Upsilon^V_1, \ldots, \Upsilon^V_t$. Let $\delta^v(t) = \delta/(12KSAt^2)$. Now,

$$\mathbb{P}\left[|\hat{v}_{m,k}(s,a) - v_k(s,a)| \leq \sqrt{\frac{2\hat{\sigma}^2_{v,m,k}\log(1/\delta^v(N^+_m(s,a)))}{N^+_m(s,a)}} + \frac{3\log(1/\delta^v(N^+_m(s,a)))}{N^+_m(s,a)} \forall\, m\right]$$

$$\geq \mathbb{P}\left[\left|\hat{\Upsilon}^V_t - v_k(s,a)\right| \leq \sqrt{\frac{2\hat{\sigma}^2_{\Upsilon^V,t}\log(1/\delta^v(t))}{t}} + \frac{3\log(1/\delta^v(t))}{t} \text{ for all } t \in [T]\right] \qquad (18)$$

$$\geq 1 - 3\sum_{t=1}^{T}\delta^v(t) = 1 - \frac{\delta}{4KSA}\sum_{t=1}^{T}\frac{1}{t^2} \geq 1 - \frac{\delta}{2KSA}. \qquad (19)$$

Step (18) is by applying a union bound over all possible values of $N^+_m(s,a)$s. Step (19) is by applying Theorem C.1. Finally, note that $\hat{\sigma}^2_{v,m,k} \leq \hat{v}_{m,k}(s,a)$, since $V(s_t,a_t) \in [0,1]$. Putting in the definition of $\delta^v(t)$ yields

$$\mathsf{rad}^v_{m,k}(s,a) \geq \sqrt{\frac{2\hat{\sigma}^2_{v,m,k}\log(1/\delta^v(N^+_m(s,a)))}{N^+_m(s,a)}} + \frac{3\log(1/\delta^v(N^+_m(s,a)))}{N^+_m(s,a)}.$$

Altogether, the required inequality for $\mathcal{E}^v$ is shown.

Next, we analyze the event $\mathcal{E}^p$ by in a similar way. Consider fixed states $s', s \in \mathcal{S}$ and a fixed action $a \in \mathcal{A}_s$. We assert that

$$\mathbb{P}\left[|\hat{p}_m(s'|s,a) - p(s'|s,a)| \leq \mathsf{rad}^p_m(s'|s,a) \text{ for all } m\right] \geq 1 - \frac{\delta}{2S^2A}. \qquad (20)$$

Assuming inequality (20), the bound $\Pr[\mathcal{E}^p] \geq 1 - \delta/2$ is established by taking a union bound over $s', s \in \mathcal{S}$ and $a \in \mathcal{A}_s$. Let $\Upsilon^p_1, \ldots, \Upsilon^p_T$ be $T$ i.i.d. Bernoulli random variables with the common mean $p(s'|s,a)$. For each $t \in [T]$, denote $\hat{\Upsilon}^p_t, \hat{\sigma}^2_{\Upsilon^p,t}$ respectively as the sample mean and sample variance of $\Upsilon^p_1, \ldots \Upsilon^p_t$. In addition, let $\delta^p(t) = \delta/(12S^2At^2)$. We have

$$\mathbb{P}\left[|\hat{p}_m(s'|s,a) - p(s'|s,a)| \leq \mathsf{rad}^p_m(s'|s,a) \text{ for all } m\right]$$

$$\geq \mathbb{P}\left[\left|\hat{\Upsilon}^p_t - p(s'|s,a)\right| \leq \sqrt{\frac{2\hat{\sigma}^2_{\Upsilon^p,t}\log(1/\delta^p(t))}{t}} + \frac{3\log(1/\delta^p(t))}{t} \text{ for all } t \in [T]\right]$$

$$\geq 1 - 3\sum_{t=1}^{T}\delta^p(t) = 1 - \frac{\delta}{4S^2A}\sum_{t=1}^{T}\frac{1}{t^2} \geq 1 - \frac{\delta}{2S^2A}.$$

Hence, the Lemma is proved. □

## C.2 The Remaining Proof of Lemma 4.3

In this Appendix, we continue with the proof of inequality (12), which states that

$$|\mathcal{M}_\Psi(T)| \leq M_\Psi(T) := 1 + (\sqrt{K}Q/2L_0) + 4\sqrt{2L_0T/(\sqrt{K}Q)}.$$

Now, recall from inequality (14) that we arrive at

$$\frac{(\tau(m_j+1) - \tau(m_{j-1}+1))^2}{\tau(m_{j-1}+1)} > \frac{\sqrt{K}Q}{2L_0},$$

which means

$$\tau(m_j+1) \geq \tau(m_{j-1}+1) + \sqrt{\frac{\sqrt{K}Q}{2L_0} \cdot \tau(m_{j-1}+1)}. \qquad (21)$$

Equipped with inequality (21) we apply the following technical Claim:

**Claim C.2.** *Let $C > 0$. Suppose the sequence $\{\rho_j\}_{j=1}^{\infty}$ satisfies $\rho_1 \geq C^2$, and $\rho_{j+1} \geq \rho_j + C\sqrt{\rho_j}$. Then for all $j \geq 1$ we have*

$$\rho_j \geq \frac{C^2(j-1)^2}{16}. \tag{22}$$

We apply the Claim with $C = \sqrt{\sqrt{K}Q/(2L_0)}$, and with the sequence $\rho_j = \tau(m_{\lceil C^2 \rceil + j})$ for $j = 1, 2, \ldots$, then we arrive at

$$\tau(m_{\lceil \sqrt{K}Q/(2L_0) \rceil + j}) \geq \frac{\sqrt{K}Q(j-1)^2}{32L_0}. \tag{23}$$

Finally, recall the random variable $n_\Psi = |\mathcal{M}|$. If $n_\Psi \leq \sqrt{K}Q/(2L_0)$, then clearly (12) is established. Otherwise, we put $j = n_\Psi - \lceil \sqrt{K}Q/(2L_0) \rceil - 1$ in inequality (23), which gives

$$T \geq \tau(m_{n_\Psi}) \geq \tau\left(m_{\lceil C^2 \rceil + [n_\Psi - \lceil C^2 \rceil - 1]} + 1\right) \geq \frac{\sqrt{K}Q}{32L_0} \cdot \left(n_\Psi - \frac{\sqrt{K}Q}{32L_0} - 1\right)^2. \tag{24}$$

Finally, unravelling the bound (24) gives the required upper bound. To complete the argument, we return to the proof of Claim C.2:

*Proof of Claim C.2.* We prove the required inequality (22) by induction on $j$. Inequality (22) is clearly true for $j = 1, 2$. Now, suppose that inequality (22) is true for some $j \geq 2$, we aim to show that it is also true for $j + 1$:

$$\rho_{j+1} \geq (C/4)^2(j-1)^2 + C \cdot (C/4)(j-1)$$
$$= (C/4)^2 \cdot \left[(j-1)^2 + 4(j-1)\right] \geq (C/4)^2 \cdot j^2.$$

Altogether, the claim is proved. □

## C.3 Proof of Proposition 4.2, which Decomposes the Regret

**Proposition 4.2.** *Consider an execution of TFW-UCRL2 on a communicating instance with diameter $D$. For each $T \in \mathbb{N}$, suppose that there is a deterministic constant $M(T)$ s.t. $\Pr[m(T) \leq M(T)] = 1$. Conditioned on events $\mathcal{E}^v, \mathcal{E}^p$, with probability at least $1 - O(\delta)$ we have*

$$\sum_{t=1}^{T} \theta_t^{\top}[v^* - V_t(s_t, a_t)] = \tilde{O}\left((Q\sqrt{K} + \bar{L}D)M(T)\right) + \tilde{O}\left(\bar{L}(D+1)\sqrt{\Gamma SAT}\right).$$

The proof of Proposition 4.2 is quite long, thus we divide the proof into several Appendix Sections. In the following, we provide a roadmap for the proof, and provide the statements of Lemmas C.3, C.4, C.6, C.7 and Claim C.5 that constitute the main part of the proof. These Lemmas and Claim are proved in the subsequent Appendix Sections, after we provide the auxiliary results in Appendix Section C.3.1 for the analysis.

Lemma C.3 is the Decomposition Lemma that breaks down $\theta_t^{\top}[v^* - V_t(s_t, a_t)]$ into manageable components for further analysis:

**Lemma C.3.** *Consider an execution of TFW-UCRL2 on a communicating instance. Let $t$ be a time index, and let $m$ be the episode index such that $\tau(m) \leq t < \tau(m+1)$. In addition, let $(\tilde{\phi}_m, \tilde{\gamma}_m)$ be the auxiliary output of EVI, which is applied at the start of episode $m$. Conditional on events $\mathcal{E}^v, \mathcal{E}^p$, the following inequality holds:*

$$\theta_t^{\top}[v^* - V_t(s_t, a_t)] \leq (\clubsuit_t) + (\diamondsuit_t) + (\heartsuit_t) + (\spadesuit_t) + (\P_t),$$

*where $v^* = \sum_{s \in \mathcal{S}, a \in \mathcal{A}_s} v(s, a)x^*(s, a)$ with $x^*$ optimal for $(\mathsf{P}_\mathcal{M})$, and*

$$(\clubsuit_t) := \left[\theta_{\tau(m)} - \theta_t\right]^{\top} V_t(s_t, a_t), \qquad\qquad (\diamondsuit_t) := \tilde{r}_m(s_t, a_t) - \theta_{\tau(m)}^{\top} V_t(s_t, a_t), \tag{25}$$

$$(\heartsuit_t) := \left[\theta_t - \theta_{\tau(m)}\right]^{\top} v^*, \qquad (\spadesuit_t) := \max_{\bar{p} \in H_m^p(s_t, a_t)} \left\{\sum_{s' \in \mathcal{S}} \tilde{\gamma}_m(s')\bar{p}(s')\right\} - \tilde{\gamma}_m(s_t), \tag{26}$$

$$(\P_t) := 1/\sqrt{\tau(m)}. \tag{27}$$

A proof of Lemma C.3 is provided in Appendix C.3.2. The proof is based on relating $(\tilde{\phi}_m, \tilde{\gamma}_m)$, the auxiliary dual variable output by EVI, to the dual of $(\mathrm{P}_{\mathcal{M}})$ with linearized reward $\tilde{r}_m$. The error terms $(25 - 27)$ account for the shortfall of the global reward collected by TFW-UCRL2, compared to the offline reward. Next, we start the bounding of the error terms by first bounding $(\clubsuit, \heartsuit)$, which account for the error by the delay of gradient updates:

**Lemma C.4.** *Suppose that gradient threshold $Q > 0$, and $\Pr[m(T) \leq M(T)] = 1$ for some deterministic constant $M(T)$. With probability 1,*

$$\sum_{t=1}^{T}(\clubsuit_t) \leq Q\sqrt{K}M(T), \qquad \sum_{t=1}^{T}(\heartsuit_t) \leq Q\sqrt{K}M(T).$$

Lemma C.4 is proved in Appendix C.3.3. We next bound $(\P, \heartsuit, \spadesuit)$, similarly to the styles in [28, 22], but with important changes to adapt to our episode schedule. The error terms $(\P, \heartsuit, \spadesuit)$ account for the error due to optimistic exploration, and the term $(\spadesuit)$ also penalizes for episode changes, which lead to sub-optimality due to the switches in stationary policies and disrupt learning.

**Claim C.5.** *With certainty, we have $\sum_{t=1}^{T}(\P_t) \leq \left(\sqrt{2}+1\right)\sqrt{T}$.*

**Lemma C.6.** *Conditional on event $\mathcal{E}^v$, with probability at least $1 - \delta$ we have:*

$$\sum_{t=1}^{T}(\Diamond_t) = \tilde{O}\left(\bar{L}\sqrt{SAT}\right).$$

**Lemma C.7.** *Suppose that $\mathcal{M}$ is communicating with diameter $D$, and $\mathbb{P}[m(T) \leq M(T)] = 1$ for some deterministic constant $M(T)$. Conditional on event $\mathcal{E}^p$, with probability at least $1 - \delta$ we have*

$$\sum_{t=1}^{T}(\spadesuit_t) = \tilde{O}\left(\bar{L}D \cdot M(T)\right) + \tilde{O}\left(\bar{L} \cdot D\sqrt{\Gamma SAT}\right).$$

The proofs of Claim C.5, Lemmas C.6, C.7 are provided in Appendices C.3.4, C.3.5, C.3.6 respectively. Altogether, Proposition 4.2 is proved by summing the bounds for $(\clubsuit, \Diamond, \heartsuit, \spadesuit, \P)$. ∎

### C.3.1 Auxiliary results for analyzing TFW-UCRL2

In order to prove Lemmas C.3, C.4, C.6, C.7 and Claim C.5, we need the following auxiliary results. First, we state the Hoeffding inequality for analyzing the dynamics of the online processes.

**Theorem C.8** ([26])**.** *Let random variables $X_1, \ldots, X_T$ constitute a martingale difference sequence w.r.t. a filtration $\{\mathcal{F}_t\}_{t=1}^{T}$, that is, $\mathbb{E}[X_t|\mathcal{F}_{t-1}] = 0$ for all $1 \leq t \leq T$. Also, suppose that $|X_t| \leq B$ almost surely for all $t$. Then the following inequality holds for any $0 < \delta < 1$:*

$$\Pr\left[\frac{1}{T}\sum_{t=1}^{T}X_t \leq B\sqrt{\frac{2\log(1/\delta)}{T}}\right] \geq 1 - \delta.$$ ∎

Next, we present auxiliary results, mostly from [28]. Theorem C.9 is useful for analyzing EVI. Lemmas C.10, C.11 and Claim C.12 are useful for proving the convergence of TFW-UCRL2.

**Theorem C.9** ([28])**.** *Consider applying EVI (Algorithm 2) with input $(\tilde{r}, H^p; \epsilon)$, where the underlying transition kernel $p$ of lies in $H^p$, and the underlying instance is communicating with diameter $D$. Then (i) EVI$(\tilde{r}, H^p; \epsilon)$ terminates in finite time, (ii) the output dual variables $\tilde{\gamma}$ satisfies $\max_{s \in \mathcal{S}} \tilde{\gamma}_s - \min_{s \in \mathcal{S}} \tilde{\gamma}_s \leq D \cdot \max_{s, \in \mathcal{S}, a \in \mathcal{A}} |\tilde{r}(s, a)|$.* ∎

**Lemma C.10** (Lemma 19 in [28])**.** *For any sequence of numbers $z_1, \ldots, z_n$ with $0 \leq z_m \leq Z_{m-1} := \max\{1, \sum_{i=1}^{m-1} z_i\}$, we have*

$$\sum_{m=1}^{n} \frac{z_m}{\sqrt{Z_{m-1}}} \leq \left(\sqrt{2}+1\right)\sqrt{Z_n}.$$ ∎

**Lemma C.11** ([28])**.** *The following inequality holds with certainty:*

$$\sum_{t=1}^{T} \frac{1}{\sqrt{N_{m(t)}^{+}(s_t, a_t)}} \leq \left(\sqrt{2}+1\right)\sqrt{SAT}.$$ ∎

**Claim C.12.** *The following inequality holds with certainty:*

$$\sum_{t=1}^{T} \frac{1}{N_{m(t)}^{+}(s_t, a_t)} \le SA \left(1 + 2 \log T\right).$$

*Proof of Claim C.12.* To start the proof, first denote $\nu'_{m(T)}(s,a) = \sum_{t=\tau(m(T))}^{T} 1((s_t, a_t) = (s,a))$. Essentially $\nu'_{m(T)}(s,a)$ is $\nu_{m(T)}(s,a)$ capped at the end of time step $T$. In addition, denote $N_{m(T)+1}^{+'}(s,a) = \sum_{t=1}^{T} 1((s_t, a_t) = (s,a))$. Similar to $\nu'_{m(T)}(s,a)$, $N_{m(T)+1}^{+'}(s,a)$ denotes the version of $N_{m(T)+1}^{+}(s,a)$ capped at the end of time step $T$. Now,

$$\sum_{t=1}^{T} \frac{1}{N_{m(t)}^{+}(s_t, a_t)} = \sum_{m=1}^{m(T)-1} \sum_{s \in \mathcal{S}} \sum_{a \in \mathcal{A}_s} \frac{\nu_m(s,a)}{N_m^{+}(s,a)} + \sum_{s \in \mathcal{S}} \sum_{a \in \mathcal{A}_s} \frac{\nu'_{m(T)}(s,a)}{N_{m(T)}^{+}(s,a)}.$$

Now, for every state-action pair $s, a$, we assert that

$$\sum_{m=1}^{m(T)-1} \frac{\nu_m(s,a)}{N_m^{+}(s,a)} + \frac{\nu'_{m(T)}(s,a)}{N_{m(T)}^{+}(s,a)} \le 1 + 2 \log \left(N_{m(T)+1}^{+'}(s,a)\right). \tag{28}$$

Indeed, the asserted inequality can be proved by drawing the following general fact: For any sequence of numbers $z_1, \ldots, z_n$ with $0 \le z_m \le Z_{m-1} := \max\{1, \sum_{i=1}^{m-1} z_i\}$, we have

$$\sum_{m=1}^{n} \frac{z_m}{Z_{m-1}} \le 1 + 2 \log Z_n. \tag{29}$$

We prove the inequality (29) by induction on $n$. The case for $n = 1$ is clearly true. Now, suppose the inequality is true for $n$. Then it is also true for $n + 1$, since

$$\sum_{m=1}^{n+1} \frac{z_m}{Z_{m-1}} \le 1 + 2 \log Z_n + \frac{z_{n+1}}{Z_n} \le 1 + 2 \log Z_n + 2 \log \left(1 + \frac{z_{n+1}}{Z_n}\right) = 1 + 2 \log Z_{n+1},$$

where we use the fact that $x \le 2 \log(1 + x)$ for $x \in [0, 1]$. Hence, the induction is established and the (29) is proved for general $n$.

Given (29) for general $n$, we can readily establish (28) by applying (29) with $n = m(T)$, $z_m = \nu_m(s,a)$ for $1 \le m \le n - 1$ and $z_n = \nu'_n(s,a)$. Altogether, noting that $N_{m(T)+1}^{+'}(s,a) \le T$, we achieve the required inequality. $\qquad\square$

### C.3.2 Proof of Lemma C.3, which decomposes the regret

First, by the definitions of $(\clubsuit_t), (\diamondsuit_t)$, it is clear that

$$\theta_t^\top V_t(s_t, a_t) = \tilde{r}_m(s_t, a_t) - [(\clubsuit_t) + (\diamondsuit_t)].$$

Thus, it suffices to show that

$$\tilde{r}_m(s_t, a_t) \ge \theta_t^\top \sum_{s \in \mathcal{S}, a \in \mathcal{A}_s} v(s,a) x^*(s,a) - [(\heartsuit_t) + (\spadesuit_t) + (\P_t)]. \tag{30}$$

To prove (30), we first focus on the application EVI. By Assumption 2.1 and by assuming the event $\mathcal{E}^p$, we know that the oracle terminates in finite time, by virtue of item (i) in Theorem C.9. Thus, the output policy $\tilde{\pi}_m$ and the auxiliary output dual variables $(\tilde{\phi}_m, \tilde{\gamma}_m)$ (Line 9 in EVI, Algorithm 2) are well-defined. Now, we assert that

$$\tilde{r}_m(s_t, a_t) \ge \tilde{\phi}_m - [(\spadesuit_t) + (\P_t)]. \tag{31}$$

To show (31), we let $\tilde{u}_{\iota+1}, \tilde{u}_\iota \in \mathbb{R}^\mathcal{S}$ respectively be the terminating and the penultimate VI records, when $\text{EVI}(\tilde{r}_m, H_p^m, 1/\sqrt{\tau(m)})$ is applied. Now, we have

$$\tilde{\phi}_m - (\P_t) = \max_{s \in \mathcal{S}} \{\tilde{u}_{\iota+1}(s) - \tilde{u}_\iota(s)\} - \frac{1}{\sqrt{\tau(m)}} \tag{32}$$

$$\leq \min_{s \in \mathcal{S}} \{\tilde{u}_{\iota+1}(s) - \tilde{u}_{\iota}(s)\} \tag{33}$$

$$\leq \tilde{u}_{\iota+1}(s_t) - \tilde{u}_{\iota}(s_t)$$

$$= \max_{a \in \mathcal{A}_{s_t}} \left\{ \tilde{r}_m(s_t, a) + \max_{\bar{p} \in H_m^p(s_t, a)} \left\{ \sum_{s' \in \mathcal{S}} \tilde{u}_{\iota}(s') \bar{p}(s') \right\} \right\} - \tilde{u}_{\iota}(s_t)$$

$$= \tilde{r}_m(s_t, a_t) + \max_{\bar{p} \in H_m^p(s_t, a_t)} \left\{ \sum_{s' \in \mathcal{S}} \tilde{u}_{\iota}(s') \bar{p}(s') \right\} - \tilde{u}_{\iota}(s_t) \tag{34}$$

$$= \tilde{r}_m(s_t, a_t) + (\spadesuit_t), \tag{35}$$

where step (32) is by the definition of $\tilde{\phi}_m$, step (33) is by the terminating condition of EVI, and step (34) is by the definition of $\tilde{\pi}_m$, and step (35) is by the definition of $\tilde{\gamma}_m$.

In order to prove the inequality (30) and complete the proof of the Lemma, it suffices to show

$$\tilde{\phi}_m \geq \theta_t^\top \sum_{s \in \mathcal{S}, a \in \mathcal{A}_s} v(s, a) x^*(s, a) - (\heartsuit_t). \tag{36}$$

To this end, we first claim that the auxiliary output dual variables $(\tilde{\phi}_m, \tilde{\gamma}_m)$ are feasible to the following linear program (lin-D$_m$):

(lin-D$_m$):  $\min \phi$

$$\text{s.t. } \phi + \gamma(s) \geq \tilde{r}_m(s, a) + \sum_{s' \in \mathcal{S}} p(s'|s, a) \gamma(s') \quad \forall s \in \mathcal{S}, a \in \mathcal{A}_s$$

$$\phi, \gamma(s) \text{ free} \quad \forall s \in \mathcal{S}.$$

Indeed, for any $s \in \mathcal{S}, a \in \mathcal{A}_s$, we have

$$\tilde{\phi}_m + \tilde{\gamma}_m(s) \geq \tilde{u}_{\iota+1}(s) - \tilde{u}_{\iota}(s) + \tilde{u}_{\iota}(s) = \tilde{u}_{\iota+1}(s)$$

$$\geq \tilde{r}_m(s, a) + \max_{\bar{p} \in H_m^p(s, a)} \left\{ \sum_{s' \in \mathcal{S}} \tilde{u}_{\iota}(s') \bar{p}(s') \right\} \tag{38}$$

$$\geq \tilde{r}_m(s, a) + \sum_{s' \in \mathcal{S}} \tilde{u}_{\iota}(s') p(s'|s, a) = \tilde{r}_m(s, a) + \sum_{s' \in \mathcal{S}} \tilde{\gamma}_m(s') p(s'|s, a),$$

where step (38) is by the assumption that $p \in H_m^p$, since we condition on the event $\mathcal{E}^p$. Therefore, we have $\tilde{\phi}_m \geq \text{opt}(\text{lin-D}_m) = \text{opt}(\text{lin-P}_m)$, where the linear program

(lin-P$_m$):  $\max \sum_{s \in \mathcal{S}, a \in \mathcal{A}_s} \tilde{r}_m(s, a) x(s, a)$

$$\text{s.t. } \sum_{a \in \mathcal{A}_s} x(s, a) = \sum_{s' \in \mathcal{S}, a' \in \mathcal{A}_{s'}} P(s|s', a') x(s', a') \quad \forall s \in \mathcal{S}$$

$$\sum_{s \in \mathcal{S}, a \in \mathcal{A}_s} x(s, a) = 1$$

$$x(s, a) \geq 0 \quad \forall s \in \mathcal{S}, a \in \mathcal{A}_s$$

is a dual formulation of (lin-D$_m$). The optimal solution $x^*$ of the offline benchmark problem (P$_\mathcal{M}$) is feasible to the problem (lin-P$_m$), since both (P$_\mathcal{M}$), (lin-P$_m$) have the same feasible region.

Finally, we prove the inequality (36), and hence completing the proof of the Lemma. In the following derivation, we denote $\tilde{v}_m(s, a)$ as an optimal solution to the optimization problem:

$$\tilde{r}_m(s, a) = \max_{\bar{v}(s, a) \in H_m^v(s, a)} \theta_{\tau(m)}^\top \bar{v}(s, a), \tag{40}$$

which is solved for computing $\tilde{r}_m(s, a)$ in Line 6 in TFW-UCRL2, Algorithm 1:

$$\tilde{\phi}_m \geq \sum_{s \in \mathcal{S}, a \in \mathcal{A}_s} \tilde{r}_m(s, a) x^*(s, a)$$

$$= \theta_{\tau(m)}^\top \sum_{s\in\mathcal{S}, a\in\mathcal{A}_s} \tilde{v}_m(s,a) x^*(s,a)$$

$$= \sum_{s\in\mathcal{S}, a\in\mathcal{A}_s} x^*(s,a) \left[ \theta_{\tau(m)}^\top \tilde{v}_m(s,a) - \theta_{\tau(m)}^\top v(s,a) \right]$$

$$+ \left[ \theta_{\tau(m)} - \theta_t \right]^\top \sum_{s\in\mathcal{S}, a\in\mathcal{A}_s} v(s,a) x^*(s,a) + \theta_t^\top \sum_{s\in\mathcal{S}, a\in\mathcal{A}_s} v(s,a) x^*(s,a) \quad (41)$$

$$\geq -(\heartsuit_t) + \theta_t^\top \sum_{s\in\mathcal{S}, a\in\mathcal{A}_s} v(s,a) x^*(s,a),$$

where step (41) holds, since we condition on the event $\mathcal{E}^v$, which ensures that $\theta_{\tau(m)}^\top \tilde{v}_m(s,a) - \theta_{\tau(m)}^\top v(s,a) \geq 0$ for each $s \in \mathcal{S}, a \in \mathcal{A}_s$. Therefore, the first sum in (41) is non-negative, hence the step is justified. Altogether, inequality (36) is shown, and the Lemma is proved. ∎

### C.3.3 Proof of Lemma C.4, which bounds $(\clubsuit, \heartsuit)$

Now,

$$\sum_{t=1}^{T}(\clubsuit_t) = \sum_{m=1}^{m(T)-1} \sum_{t=\tau(m)}^{\tau(m+1)-1} \left[\theta_{\tau(m)} - \theta_t\right]^\top V_t(s_t, a_t) + \sum_{t=\tau(m(T))}^{T} \left[\theta_{\tau(m)} - \theta_t\right]^\top V_t(s_t, a_t)$$

$$\leq \sum_{m=1}^{m(T)-1} \sum_{t=\tau(m)}^{\tau(m+1)-1} \left\|\theta_{\tau(m)} - \theta_t\right\|_2 \left\|V_t(s_t, a_t)\right\|_2 + \sum_{t=\tau(m(T))}^{T} \left\|\theta_t - \theta_{\tau(m)}\right\|_2 \left\|V_t(s_t, a_t)\right\|_2$$

$$\tag{42}$$

$$\leq \sum_{m=1}^{m(T)-1} Q\sqrt{K} + Q\sqrt{K} \quad \text{w.p. } 1 \tag{43}$$

$$= Q\sqrt{K} \cdot M(T).$$

Step (42) is by the triangle inqeuality and the Cauchy-Scwhartz inequality. Step (43) is by our terminating criteria, which require $\Psi \leq Q$ for each episode, as well as the model assumption that $V(s,a) \in [0,1]^K$. Similar to the above, we also have:

$$\sum_{t=1}^{T}(\heartsuit_t) = \left\{ \sum_{m=1}^{m(T)-1} \sum_{t=\tau(m)}^{\tau(m+1)-1} \left[\theta_t - \theta_{\tau(m)}\right] + \sum_{t=\tau(m(T))}^{T} \left[\theta_t - \theta_{\tau(m)}\right] \right\}^\top \sum_{s\in\mathcal{S}, a\in\mathcal{A}_s} v(s,a) x^*(s,a)$$

$$\leq \left\{ \sum_{m=1}^{m(T)-1} \sum_{t=\tau(m)}^{\tau(m+1)-1} \left\|\theta_t - \theta_{\tau(m)}\right\|_2 + \sum_{t=\tau(m(T))}^{T} \left\|\theta_t - \theta_{\tau(m)}\right\|_2 \right\} \left\| \sum_{s\in\mathcal{S}, a\in\mathcal{A}_s} v(s,a) x^*(s,a) \right\|_2$$

$$\leq \sum_{m=1}^{m(T)-1} Q\sqrt{K} + Q\sqrt{K} \quad \text{w.p. } 1$$

$$= Q\sqrt{K} \cdot M(T).$$

Altogether, the Lemma is proved. ∎

### C.3.4 Proof of Claim C.5, which bounds $(\P)$

The proof uses Lemma C.10. Let's apply $n = m(T)$, as well as

$$z_m = \begin{cases} \tau(m+1) - \tau(m) & \text{if } 1 \leq m < m(T) \\ T - \tau(m) & \text{if } m = m(T) \end{cases},$$

where we set $\tau(0) = 0$. Now, $Z_0 = 1$, $Z_m = \tau(m)$ for $1 \leq m < m(T)$, and $Z_{m(T)} = T$. Therefore,

$$\sum_{t=1}^{T}(\P_t) = \sum_{m=1}^{m(T)-1} \sum_{t=\tau(m)}^{\tau(m+1)-1} \frac{1}{\sqrt{\tau(m)}} + \sum_{t=\tau(m(T))}^{T} \frac{1}{\sqrt{\tau(m(T))}}$$

$$= \sum_{m=1}^{n} \frac{z_m}{\sqrt{Z_{m-1}}} \le \left( \sqrt{2} + 1 \right) \sqrt{Z_{m(T)}} = \left( \sqrt{2} + 1 \right) \sqrt{T}.$$

Hence the claim is proved. ∎

### C.3.5 Proof of Lemma C.6, which bounds ($\Diamond$)

The proof of the Lemma uses the Azuma-Hoeffding inequality in Theorem C.8, as well as Lemma C.11 by [28] and Claim C.12.

To start the proof, we define $\tilde{v}_m(s, a)$ and $m(t)$. We express $\tilde{r}_m(s, a) = \theta_{\tau(m)}^\top \tilde{v}_m(s, a)$, where $\tilde{v}_m(s, a)$ is an optimal solution to the optimization problem (40) in the proof of the Decomposition Lemma, Lemma C.3. For each $t$, we define $m(t)$ to be the episode index such that $\tau(m(t)) \le t < \tau(m(t) + 1) - 1$. We first decompose $\sum_{t=1}^{T}(\Diamond_t)$ as follows:

$$\sum_{t=1}^{T}(\Diamond_t) \le \sum_{t=1}^{T} \tilde{r}_{m(t)}(s_t, a_t) - \theta_{\tau(m(t))}^\top V_t(s_t, a_t)$$

$$= \underbrace{\sum_{t=1}^{T} \theta_{\tau(m(t))}^\top \left[ \tilde{v}_{m(t)}(s_t, a_t) - v(s_t, a_t) \right]}_{(\dagger_v)} + \underbrace{\sum_{t=1}^{T} \theta_{\tau(m(t))}^\top \left[ v(s_t, a_t) - V_t(s_t, a_t) \right]}_{(\ddagger_v)}.$$

We bound the sums $(\dagger_v, \ddagger_v)$ as follows:

**Bounding** $(\dagger_v)$. We bound this term by invoking the confidence bounds asserted by the event $\mathcal{E}^v$. Define the notation $(\text{log-}v) := \log(12KSAT^2/\delta)$. We have

$$(\dagger_v) = \sum_{t=1}^{T} \theta_{\tau(m(t))}^\top \left[ \tilde{v}_{m(t)}(s_t, a_t) - \hat{v}_{m(t)}(s_t, a_t) + \hat{v}_{m(t)}(s_t, a_t) - v(s_t, a_t) \right]$$

$$\le \sum_{t=1}^{T} \left\| \theta_{\tau(m(t))} \right\|_2 \left[ \left\| \tilde{v}_{m(t)}(s_t, a_t) - \hat{v}_{m(t)}(s_t, a_t) \right\|_2 + \left\| \hat{v}_{m(t)}(s_t, a_t) - v(s_t, a_t) \right\|_2 \right] \quad (44)$$

$$\le 2 \frac{\bar{L}}{\sqrt{K}} \sum_{t=1}^{T} \left\| (\text{rad}_{m(t),k}^v(s_t, a_t))_{k=1}^{K} \right\|_2 \quad (45)$$

$$\le 4\bar{L} \left[ \sqrt{(\text{log-}v)} \cdot \sum_{t=1}^{T} \frac{1}{\sqrt{N_{m(t)}^+(s_t, a_t)}} + 3 \cdot (\text{log-}v) \cdot \sum_{t=1}^{T} \frac{1}{N_{m(t)}^+(s_t, a_t)} \right] \quad (46)$$

$$\le 4\bar{L} \left[ \left( \sqrt{2} + 1 \right) \sqrt{SAT \cdot (\text{log-}v)} + 3 \cdot (\text{log-}v) \cdot SA (1 + 2\log T) \right].$$

Step (44) is by the Cauchy-Schwartz inequality, step (45) is by the assumption that the event $\mathcal{E}^v$ holds, as well as the fact that $\|\theta_{\tau(m(t))}\|_2 \le \bar{L}/\sqrt{K}$, since $g$ is $(\bar{L}/\sqrt{K})-$Lipschitz w.r.t. $\|\cdot\|_2$. Step (46) is by Lemma C.11 and Claim C.12, as well as $(\text{log-}v) \ge (\text{log-}v)_m$ for all $m$.

**Bounding** $(\ddagger_v)$. Consider random variable $X_t = \theta_{\tau(m(t))}^\top \left[ v(s_t, a_t) - V_t(s_t, a_t) \right]$ and filtration $\mathcal{F}_t = \sigma(\{s_t, a_t, V_t(s_t, a_t), \theta_{t+1}\}_{\tau=1}^{t})$. Now, $|X_t| \le \bar{L}$, $X_t$ is $\mathcal{F}_t$-measurable with $\mathbf{E}[X_t | \mathcal{F}_{t-1}] = 0$. Thus, we apply Theorem C.8 to conclude that, with probability $\ge 1 - \delta$,

$$(\ddagger_v) \le \bar{L}\sqrt{2T \log(1/\delta)}.$$

Altogether, we have, with probability at least $1 - \delta$,

$$\sum_{t=1}^{T}(\Diamond_t) \le \bar{L} \left[ \left( 5\sqrt{2} + 4 \right) \sqrt{\sum_{s \in \mathcal{S}} |\mathcal{A}_s| T \cdot (\text{log-}v)} + 12 \cdot (\text{log-}v) \cdot \sum_{s \in \mathcal{S}} |\mathcal{A}_s| \log T \right] \quad (47)$$

$$= O \left( \bar{L}\sqrt{SAT \log \frac{KSAT}{\delta}} + \bar{L}SA \log^2 \frac{KSAT}{\delta} \right).$$

Hence, the Lemma is proved. ∎

### C.3.6 Proof of Lemma C.7, which bounds ($\spadesuit$)

First, observe that by item (ii) in Theorem C.9, we have

$$\max_{s \in \mathcal{S}} \{\tilde{\gamma}_m(s)\} - \min_{s \in \mathcal{S}} \{\tilde{\gamma}_m(s)\} \leq D \cdot \max_{s,a} \max_{w \in H_m^v(s,a)} \theta_m^\top w \leq \bar{L}D$$

for all $m$. The first inequality is by part (ii) in Theorem C.9, as well as the fact that conditioned on $\mathcal{E}^p$, we have $p \in H_m^p$ for all $m$. The second inequality is by these two inequalities: $\|\theta\|_2 \leq \bar{L}/\sqrt{K}$ by the $(\bar{L}/\sqrt{K})$-Lipschitz continuity of $g$ w.r.t. $\|\cdot\|_2$, and $\|w\|_2 \leq \sqrt{K}$ for all $w \in H_m^v(s,a)$.

For each episode $m$ and state $s$, consider replacing $\tilde{\gamma}_m(s)$ by $\gamma_m(s) := \tilde{\gamma}_m(s) - \min_{s' \in \mathcal{S}} \{\tilde{\gamma}_m(s')\}$. Now, $0 \leq \max_{m,s} \{\gamma_m(s)\} \leq \bar{L}D$, and the value of each ($\spadesuit_t$) is preserved:

$$(\spadesuit_t) = \max_{\bar{p} \in H_{m(t)}^p(s_t,a_t)} \left\{ \sum_{s' \in \mathcal{S}} \tilde{\gamma}_{m(t)}(s')\bar{p}(s') \right\} - \tilde{\gamma}_{m(t)}(s_t)$$

$$= \max_{\bar{p} \in H_{m(t)}^p(s_t,a_t)} \left\{ \sum_{s' \in \mathcal{S}} \gamma_{m(t)}(s')\bar{p}(s') \right\} - \gamma_{m(t)}(s_t),$$

where $m(t)$ is the episode index such that $\tau(m(t)) \leq t < \tau(m(t)+1)$.

Consider the following decomposition:

$$\sum_{t=1}^T (\spadesuit_t) = \sum_{t=1}^T \underbrace{\left[ \max_{\bar{p} \in H_m^p(s_t,a_t)} \left\{ \sum_{s' \in \mathcal{S}} \gamma_{m(t)}(s')\bar{p}(s') \right\} - \sum_{s \in \mathcal{S}} \gamma_{m(t)}(s)p(s|s_t,a_t) \right]}_{(\dagger_p)}$$

$$+ \sum_{t=1}^T \underbrace{\left[ \sum_{s \in \mathcal{S}} \gamma_{m(t)}(s)p(s|s_t,a_t) - \gamma_{m(t)}(s_t) \right]}_{(\ddagger_p)}.$$

**Bounding ($\dagger_p$).** We proceed by unraveling $H_m^p$. Now, denote $(\text{log-}p) := \log(12S^2AT^2/\delta)$.

$$(\dagger_p) \leq \sum_{t=1}^T \left[ \max_{\bar{p} \in H_{m(t)}^p(s_t,a_t)} \left\{ \sum_{s \in \mathcal{S}} \gamma_{m(t)}(s)\bar{p}(s) \right\} - \min_{\bar{p} \in H_{m(t)}^p(s_t,a_t)} \left\{ \sum_{s \in \mathcal{S}} \gamma_{m(t)}(s)\bar{p}(s) \right\} \right]$$

$$\leq 2 \sum_{t=1}^T \sum_{s \in \mathcal{S}} \gamma_{m(t)}(s)\mathsf{rad}_{m(t)}^p(s|s_t,a_t)$$

$$\leq 2\bar{L}D \sum_{t=1}^T \sum_{s \in \mathcal{S}} \left[ \sqrt{\frac{2\hat{p}_{m(t)}(s'|s,a) \cdot (\text{log-}p)}{N_{m(t)}^+(s,a)}} + \frac{3(\text{log-}p)}{N_{m(t)}^+(s,a)} \right]$$

$$\leq 2\bar{L}D \sum_{t=1}^T \left[ \sqrt{\frac{2\Gamma \cdot (\text{log-}p)}{N_{m(t)}^+(s,a)}} + \frac{3 \cdot S(\text{log-}p)}{N_{m(t)}^+(s,a)} \right] \tag{48}$$

$$\leq 2(\sqrt{2}+1)\bar{L}D\sqrt{2\Gamma SAT \cdot (\text{log-}p)} + 6\bar{L}DS^2A(1 + 2\log T)(\text{log-}p). \tag{49}$$

We justify step (48) as follows. Now, recall $\Gamma = \max_{s \in \mathcal{S}, a \in \mathcal{A}_s} \|p(\cdot|s,a)\|_0$. With certainty, we have $\|\hat{p}_m(\cdot|s,a)\|_0 \leq \|p(\cdot|s,a)\|_0 \leq \Gamma$. Indeed, for each $s' \in \mathcal{S}$, $p(s'|s,a) = 0$ implies that $\hat{p}_m(s'|s,a) = 0$ with certainty. By the Cauchy-Schwartz inequality,

$$\sum_{s' \in \mathcal{S}} \sqrt{\hat{p}_m(s'|s,a)} = \sum_{s' \in \mathcal{S}} \sqrt{\hat{p}_m(s'|s,a) \cdot \mathbb{1}(p(s'|s,a) > 0)}$$

$$\leq \sqrt{\left[ \sum_{s' \in \mathcal{S}} \hat{p}_m(s'|s,a) \right] \left[ \sum_{s' \in \mathcal{S}} \mathbb{1}(p(s'|s,a) > 0) \right]} = \sqrt{\|p(\cdot|s,a)\|_0} = \sqrt{\Gamma}.$$

Step (49) is by Proposition C.11 and Claim C.12.

**Bounding** $(\ddagger_p)$. We analyze the term by accounting for the number of episodes:

$$(\ddagger_p) = \left[\gamma_{m(T+1)}(s_{T+1}) - \gamma_{m(1)}(s_1)\right] + \sum_{t=1}^{T}\left[\sum_{s \in \mathcal{S}}\gamma_{m(t)}(s)p(s|s_t, a_t) - \gamma_{m(t+1)}(s_{t+1})\right]$$

$$= \left[\gamma_{m(T+1)}(s_{T+1}) - \gamma_{m(1)}(s_1)\right] + \sum_{t=1}^{T}\left[\gamma_{m(t)}(s_{t+1}) - \gamma_{m(t+1)}(s_{t+1})\right]$$

$$+ \sum_{t=1}^{T}\left[\sum_{s \in \mathcal{S}}\gamma_{m(t)}(s)p(s|s_t, a_t) - \gamma_{m(t)}(s_{t+1})\right] \quad \text{w.p. } 1 \tag{50}$$

$$\leq \max_{t,s}\{\gamma_{m(t)}(s)\}(M(T) + 1) + \sum_{t=1}^{T}\left[\sum_{s \in \mathcal{S}}\gamma_{m(t)}(s)p(s|s_t, a_t) - \gamma_{m(t)}(s_{t+1})\right] \tag{51}$$

$$\leq \max_{t,s}\{\gamma_{m(t)}(s)\}(M(T) + 1) + \max_{t,s}\{\gamma_{m(t)}(s)\}\sqrt{2T\log(1/\delta)} \quad \text{w.p. } 1 - \delta$$

$$\leq \bar{L}D(M(T) + 1) + \bar{L}D\sqrt{2T\log(1/\delta)}.$$

Step (51) is shown by analyzing the second summation in (50), which is $\sum_{t=1}^{T}\gamma_{m(t)}(s_{t+1}) - \gamma_{m(t+1)}(s_{t+1})$. In the summation, at most $m(T) \leq M(T)$ summands are non-zero, and each non-zero summand is less than or equal to $\max_{t,s}\{\gamma_{m(t)}(s)\} \leq \bar{L}D$.

Combining the bounds for $(\dagger_p, \ddagger_p)$, with probability at least $1 - \delta$ we have

$$\sum_{t=1}^{T}(\spadesuit_t) \leq (2\sqrt{2} + 3)\bar{L}D\sqrt{2\Gamma SAT \cdot (\text{log-}p)} + \bar{L} \cdot D(M(T) + 1) \tag{52}$$

$$+ 6\bar{L}DS^2A(1 + 2\log T) \cdot (\text{log-}p)$$

$$= O\left(\bar{L}DM(T)\right) + O\left(\bar{L}D\sqrt{\Gamma SAT \log\frac{SAT}{\delta}} + \bar{L}DS^2A\log^2\frac{SAT}{\delta}\right).$$

Altogether, the Lemma is proved. ∎