[Reviews · NeurIPS 2019]

Reviewer 1



Summary: This paper studies a generalization of online reinforcement learning (in the infinite horizon undiscounted setting with finite state and action space and communicating MDP) where the agent aims at maximizing a certain type of concave function of the rewards (extended to global concave functions in appendix). More precisely, every time an action "a" is played in state "s", the agent receives a vector of rewards V(s,a) (instead of a scalar reward r(s,a)) and tries to maximize a concave function of the empirical average of the vectorial outcomes. This problem is very general and models a wide variety of different settings ranging from multi-objective optimization in MDPs, to maximum entropy exploration and online learning in MDPs with knapsack constraints. In section 2 the authors introduce the necessary background and formalize the notions of "optimal gain" and "regret" in this setting. Defining the "optimal gain" (called the "offline benchmark" in the paper) is not straightforward. In section 3, the authors first show how the exploration-exploitation dilemma faced by the agent is intrinsically more challenging since the optimal policy may be non-stationary (example of Fig. 1 and Claim 3).2. They then introduce a variant of UCRL2 (called TFW-UCRL2) which can efficiently balance exploration and exploitation in this setting. The algorithm mainly differs from UCRL2 by the stopping condition of episodes: an episode also stops whenever the "gradient " becomes too far from its initial value at the beginning of the episode. This prevents TFW-UCRL2 from converging to a stationary policy whenever the optimal policy is actually not stationary. The gradient is updated using a Frank-Wolfe algorithm. When the objective function is both Lipschitz and smooth, the regret of TOC-UCRL2 can be bounded by O(1/\sqrt{T}) (Theorem 3.1). A detailed proof sketch of Theorem 5 is presented in section 4 (the main steps and arguments are given). Finally, in Section 5 the authors report experiments illustrating the theoretical findings. Clarity: Overall the paper is well-written and clear. All the statements are very precise. The examples of Fig.1 are very helpful to understand the challenges of the setting (and they are well-explained). Correctness: The paper is very rigorous (both the main body and the appendix). The proofs are very detailed and still concise (and thus easy to read and verify). The derivation of the regret proof of Theorem 3.1 is very rigorous while being easy to follow. Significance: In my opinion, the contribution of this paper is quite substantial and non-trivial. The paper (including the appendices) makes a comprehensive and insightful analysis of the problem addressed in the paper. In my opinion, the problem addressed is very relevant for the community (the setting and results are very general).

Reviewer 2



My main concerns regarding the paper are the follows: 1) the main part of the paper is really hard to follow. There are four main building blocks of the proposed algorithm: UCRL, Frank-Wolf, EVI and GTP. Based on Algorithm 1, I believe that I could figure out how the Frank-Wolf and EVI are used in the UCRL framework, but I could not find out how GTP is implemented and what its role is. I guess it a mechanism to avoid the low gradient states, but I don't see how 2) The aggregation function is not well-motivated. I don't see that it would preserve the Pareto dominance relation which is a key notion in multi-objective optimization. 3) Can you please explain me any real-world problem which might be modeled by the proposed setup. Only synthetic experiments are presented in the experimental study. 4) The technical contribution seems marginal. Tremendous work is presented in the appendix, but if I look at Lemma 4.1, it seems that, thanks to the concavity of the aggregation function, the optimization problem can be handled quite similar way to the single objective case. In Eq. (11), the regret basically boils down to the error in value estimation which is very similar to the single objective case.

Reviewer 3



i. Novelty and Significance: The problem formulation seems novel as multi-objective scenarios are quite relevant in practice. I am not entirely well versed on existing works on multi-objective MDPs in RL, but authors claimed to have the first non-asymptotic sub-linear regret guarantee for the current setting unlike some prior works [28,34] which focuses on asymptotic convergence and work under more restricted assumptions on the MDPs. ii. Writing: The paper is overall well written with properly introduced notations and the claims are easy to follow. I made a high level check of the proof techniques which appears to be reasonable. iii. Numerical Experiments: This section could have been elaborated further (although more details are included in the Appendix). I would suggest moving some of the analysis of Sec. 4 to Appendix to include more experimental evaluations in the main text. In particular, the reported empirical results does not study any comparative performances with other existing methods --- I am curious to see how TFW-UCRL2 performs with respect to the proposed algorithms of [28] and [34], even under the setting of their restricted assumptions. =================== Post Rebuttal ========================= I have read and understood the rebuttal clarifications. I keep my score unchanged.

[Author Response · NeurIPS 2019]

**Reviewers # 5, 8:** Thank you for the appreciation! The best known lower bound is $\Omega(\sqrt{DSA/T})$, based on [26] for
the SO-OMDP setting. Our upper bound $\tilde{O}(D\sqrt{\Gamma SA/T})$ in Theorem 3.1 matches the $\tilde{O}(DS\sqrt{A/T})$ bound by [26]
for SO-OMDP. The best known upper bound for SO-OMDP is $\tilde{O}(c\sqrt{\Gamma SA/T})$ by [21], where $c \leq D$ is called the *span*,
a refined version of $D$ in the SO-OMDP setting. The notion of span is inapplicable to MO-OMDP.

**Reviewer # 6:** Thank your for the comments! For a better appreciation on our contributions, we clarify as follows:

Justifying the objective function $g_{\text{MO}}$ (5.1), (2.2), (2.3). We start by addressing (5.1). KPI stands for Key Performance
Index. For *Target Set Objectives*, specifying $U = \{w : w_k \geq \rho_k \forall 1 \leq k \leq K\} = \prod_{k=1}^{K}[\rho_k, \infty)$ is sufficient
for ensuring $\bar{V}_{1:T,k} \geq \rho_k$ whenever possible, thanks to the $\min_{u \in U}$ operator in (1). To see this, consider setting
$L_1 = \ldots = L_K = 0, L_0 = 1$. We claim that $g_{\text{MO}}(\bar{V}_{1:T}) = -(1/2K)\sum_{k=1}^{K}\max\{\rho_k - \bar{V}_{1:T,k}, 0\}^2$. Indeed,

$$g_{\text{MO}}(\bar{V}_{1:T}) = -\frac{1}{2K}\min_{u \in \prod_{k=1}^{K}[\rho_k, \infty)}\left\{\sum_{k=1}^{K}(\bar{V}_{1:T,k} - u_k)^2\right\} = -\frac{1}{2K}\sum_{k=1}^{K}\min_{u_k \in [\rho_k, \infty)}\left\{(\bar{V}_{1:T,k} - u_k)^2\right\}.$$

For the $k$th summand, if $\bar{V}_{1:T,k} \geq \rho_k$, the argmin is $\bar{V}_{1:T,k}$ and the summand $= 0$. Else, we have $\bar{V}_{1:T,k} < \rho_k$, the
argmin is $\rho_k$ and the summand $= (\rho_k - \bar{V}_{1:T,k})^2$. Thus, the claim is proved.

Maximizing $g_{\text{MO}}(\bar{V}_{1:T})$ is equivalent to minimizing $(1/2K)\sum_{k=1}^{K}\max\{\rho_k - \bar{V}_{1:T,k}, 0\}^2$. If the KPI $\rho$ is achievable,
then the optimal policy would generate $\bar{V}_{1:T}$ for which $\bar{V}_{1:T,k} \geq \rho_k$ for all $k$, yielding objective value $g_{\text{MO}}(\bar{V}_{1:T}) = 0$.
Otherwise, the shortfall of $\bar{V}_{1:T}$ compared to $\rho$ is minimized in the mean squared error sense.

(2.2): Any maximizer $\bar{V}_{1:T}^*$ of $g_{\text{MO}}(\bar{V}_{1:T}) = -(1/2K)\sum_{k=1}^{K}\max\{1 - \bar{V}_{1:T,k}, 0\}^2$ is Pareto-optimal. To see this, first
observe that the $\bar{V}_{1:T}$ generated by any policy satisfies $\bar{V}_{1:T} \in [0,1]^K$, since $V(s,a) \in [0,1]$ always. Suppose the
contrary that there is a $\tilde{V}_{1:T}$, where $\tilde{V}_{1:T,k} \geq \bar{V}_{1:T,k}^* \forall k$, and $\tilde{V}_{1:T,1} > \bar{V}_{1:T,1}^*$. These mean that $0 \leq 1 - \tilde{V}_{1:T,k} \leq$
$1 - \bar{V}_{1:T,k}^* \forall k$, and $0 \leq 1 - \tilde{V}_{1:T,1} < 1 - \bar{V}_{1:T,1}^*$. Consequently, $g_{\text{MO}}(\tilde{V}_{1:T}) > g_{\text{MO}}(\bar{V}_{1:T}^*)$, contradicting the maximality
of $\bar{V}_{1:T}^*$ on $g_{\text{MO}}$. Thus, $\bar{V}_{1:T}^*$ is Pareto-optimal. Altogether, $g_{\text{MO}}$ with suitably chosen $\rho, U$ captures Pareto-optimality.
Moreover, $g_{\text{MO}}$ captures the *State Space Exploration* problem, which goes beyond Pareto-optimality.

(2.3): Capturing Pareto-optimality allows us to model many real world problems. Our framework allows any smooth
concave $g$ and not just $g_{\text{MO}}$ (App. D), which captures other applications such as Maximum Entropy Exploration [23].

The design and analysis of GTP: (5.2), (2.1), (2.4). We start by addressing (5.2). For instance (1b), we claim that
$\text{opt}(P_{\mathcal{M}}) = 0$. In addition, the solution $x^*$, defined as $x^*(s^1, \mathtt{rl}) = x^*(s^2, \mathtt{ll}) = 1/2$ and $x^*(s,a) = 0$ for all
other $s, a$, is optimal to $(P_{\mathcal{M}})$. Indeed, $x^*$ is feasible to $(P_{\mathcal{M}})$ (recall $p(s^1|s^1, \mathtt{rl}) = p(s^2|s^2, \mathtt{ll}) = 1$), and that
$\sum_{s,a} v(s,a)x^*(s,a) = \binom{0}{1} * x^*(s^2, \mathtt{ll}) + \binom{1}{0} * x^*(s^1, \mathtt{rl}) = \binom{1/2}{1/2}$, and that $g_{\text{MO}}(\sum_{s,a} v(s,a)x^*(s,a)) = 0$.

The bad policy in Line 139, which causes $\bar{V}_{1:T} \approx (1/6, 1/6)^\top$, incurs $\text{Reg}(T) = 0 - (-(1/6 - 1/2)^2) = \Omega(1)$. The
$\Omega(1)$ regret is caused by the $\Theta(T)$ *implicit switching cost*, where the agent switches between $s^1, s^2$ (hence visits $s^0$) for
$\Theta(T)$ times in $T$ time steps. In an MO-OMDP instance, the implicit switching cost occurs when the agent switches
form a recurrent class to another, and visits a state that does not contribute to the objective (like $s^0$) during the switch.

(2.1): GTP (see Lines 175-177) consists of the maintenance of distance measure $\Psi$ in Line 13 in Algo 1, and the first
criterion $\Psi < Q$ in Line 9 in Algo 1. GTP keeps the implicit switching cost bounded, while balances the contributions
by $\{\bar{V}_{1:T,k}\}_{k=1}^{K}$. As said in Lines 188-193 for Fig 1b, GTP ensures the agent only switches between $s^1, s^2$ for $O(\sqrt{T})$
times in $T$ steps, and $|\bar{V}_{1:T,k} - 0.5| = O(1/\sqrt{T})$ for $k = 1, 2$, thus $\text{Reg}(T) = O(1/\sqrt{T})$. GTP reduces the implicit
switching cost from $\Theta(T)$ to $O(\sqrt{T})$, by looping at each $s^1, s^2$ for $\Theta(\sqrt{t})$ times before switching (cf. Lines 190-191).

(2.4): Lemma 4.1 follows from concentration inequalities, which are not our contributions. GTP is new, and its
design and analysis are our novel contributions. Compared to UCRL2 for SO-OMDPs, analysing TFW-UCRL2 for
MO-OMDPs requires crucial effort on bounding two costs: (i) the implicit switch cost (see Lemma 4.3) due to GTP.
(ii) there is a *delay cost* caused by GTP on the gradient updates. In Fig 1b, the delay cost is the $O(1/\sqrt{T})$ error on
$|\bar{V}_{1:T,k} - 0.5|$. The delay cost, included by eqn. (11), is discussed in Lines 244-248 and bounded by Proposition 4.2.
These switch and delay costs are not present in UCRL2, and their analyses certainly do not follow from the literature.

**Reviewer # 8:** In fact, the regret under $Q = \bar{L}/\sqrt{K}$ (where $\bar{L} = L_0 + \max_k |L_k|$) is quite close to the optimal regret
by tuning $Q$. We chose $Q = \bar{L}/\sqrt{K}$ to optimize the dependence on $\bar{L}, K$ in the regret order bound in Theorem
3.1. The regret could be improved by tuning $Q$ online, or by optimizing $Q$ in the actual regret bound. $\rho, L_0, \ldots, L_K$
parameterize the objective function $g_{\text{MO}}$, which is assumed to be fixed, while $Q$ parameterizes the algo, so we only
consider tuning $Q$. If accepted, we will conduct the suggested empirical comparisons with [26, 28, 34] in their settings.

[Meta-Review · NeurIPS 2019]

Two out of three reviewers appreciated the contributions of this paper, with one expert reviewer praising almost every aspect of the paper. On the negative side, one reviewer took issue with the proposed setting, highlighting that the utility of the proposed objective function is somewhat dubious in the general context of multi-objective decision making. I agree with this reviewer in that having "multi-objective" in the title of the paper may set the wrong expectations for some readers, and I suggest that the authors consider changing the title of the paper for its final version to avoid such misunderstandings. Furthermore, the final version should discuss the relationship between this paper and the very recent work of Rosenberg and Mansour (2019) that studies essentially the same problem in episodic MDPs. Other than these concerns, the paper is worthy of being published without major changes. Reference: Rosenberg, Aviv, and Yishay Mansour. "Online Convex Optimization in Adversarial Markov Decision Processes." International Conference on Machine Learning. 2019.